# Beyond Disagreement-based Agnostic Active Learning

**Chicheng Zhang**
University of California, San Diego
9500 Gilman Drive, La Jolla, CA 92093
`chichengzhang@ucsd.edu`

**Kamalika Chaudhuri**
University of California, San Diego
9500 Gilman Drive, La Jolla, CA 92093
`kamalika@cs.ucsd.edu`

## Abstract

We study agnostic active learning, where the goal is to learn a classifier in a pre-specified hypothesis class interactively with as few label queries as possible, while making no assumptions on the true function generating the labels. The main algorithm for this problem is *disagreement-based active learning*, which has a high label requirement. Thus a major challenge is to find an algorithm which achieves better label complexity, is consistent in an agnostic setting, and applies to general classification problems.

In this paper, we provide such an algorithm. Our solution is based on two novel contributions; first, a reduction from consistent active learning to confidence-rated prediction with guaranteed error, and second, a novel confidence-rated predictor.

## 1 Introduction

In this paper, we study *active learning* of classifiers in an agnostic setting, where no assumptions are made on the true function that generates the labels. The learner has access to a large pool of unlabelled examples, and can interactively request labels for a small subset of these; the goal is to learn an accurate classifier in a pre-specified class with as few label queries as possible. Specifically, we are given a hypothesis class $\mathcal{H}$ and a target $\epsilon$, and our aim is to find a binary classifier in $\mathcal{H}$ whose error is at most $\epsilon$ more than that of the best classifier in $\mathcal{H}$, while minimizing the number of requested labels.

There has been a large body of previous work on active learning; see the surveys by [10, 28] for overviews. The main challenge in active learning is ensuring consistency in the agnostic setting while still maintaining low label complexity. In particular, a very natural approach to active learning is to view it as a generalization of binary search [17, 9, 27]. While this strategy has been extended to several different noise models [23, 27, 26], it is generally inconsistent in the agnostic case [11].

The primary algorithm for agnostic active learning is called *disagreement-based active learning*. The main idea is as follows. A set $V_k$ of possible risk minimizers is maintained with time, and the label of an example $x$ is queried if there exist two hypotheses $h_1$ and $h_2$ in $V_k$ such that $h_1(x) \neq h_2(x)$. This algorithm is consistent in the agnostic setting [7, 2, 12, 18, 5, 19, 6, 24]; however, due to the conservative label query policy, its label requirement is high. A line of work due to [3, 4, 1] have provided algorithms that achieve better label complexity for linear classification on the uniform distribution over the unit sphere as well as log-concave distributions; however, their algorithms are limited to these specific cases, and it is unclear how to apply them more generally.

Thus, a major challenge in the agnostic active learning literature has been to find a general active learning strategy that applies to any hypothesis class and data distribution, is consistent in the agnostic case, and has a better label requirement than disagreement based active learning. This has been mentioned as an open problem by several works, such as [2, 10, 4].

In this paper, we provide such an algorithm. Our solution is based on two key contributions, which may be of independent interest. The first is a general connection between *confidence-rated predictors* and active learning. A confidence-rated predictor is one that is allowed to abstain from prediction on occasion, and as a result, can guarantee a target prediction error. Given a confidence-rated predictor with guaranteed error, we show how to to construct an active label query algorithm consistent in the agnostic setting. Our second key contribution is a novel confidence-rated predictor with guaranteed error that applies to any general classification problem. We show that our predictor is *optimal* in the realizable case, in the sense that it has the lowest abstention rate out of all predictors guaranteeing a certain error. Moreover, we show how to extend our predictor to the agnostic setting.

Combining the label query algorithm with our novel confidence-rated predictor, we get a general active learning algorithm consistent in the agnostic setting. We provide a characterization of the label complexity of our algorithm, and show that this is better than the bounds known for disagreement-based active learning in general. Finally, we show that for linear classification with respect to the uniform distribution and log-concave distributions, our bounds reduce to those of [3, 4].

## 2 Algorithm

### 2.1 The Setting

We study active learning for binary classification. Examples belong to an instance space $\mathcal{X}$, and their labels lie in a label space $\mathcal{Y} = \{-1, 1\}$; labelled examples are drawn from an underlying data distribution $D$ on $\mathcal{X} \times \mathcal{Y}$. We use $D_{\mathcal{X}}$ to denote the marginal on $D$ on $\mathcal{X}$, and $D_{Y|X}$ to denote the conditional distribution on $Y|X = x$ induced by $D$. Our algorithm has access to examples through two oracles – an example oracle $\mathcal{U}$ which returns an unlabelled example $x \in \mathcal{X}$ drawn from $D_{\mathcal{X}}$ and a labelling oracle $\mathcal{O}$ which returns the label $y$ of an input $x \in \mathcal{X}$ drawn from $D_{Y|X}$.

Given a hypothesis class $\mathcal{H}$ of VC dimension $d$, the error of any $h \in \mathcal{H}$ with respect to a data distribution $\Pi$ over $\mathcal{X} \times \mathcal{Y}$ is defined as $\mathrm{err}_{\Pi}(h) = \mathbb{P}_{(x,y)\sim\Pi}(h(x) \neq y)$. We define: $h^*(\Pi) = \mathrm{argmin}_{h\in\mathcal{H}}\mathrm{err}_{\Pi}(h)$, $\nu^*(\Pi) = \mathrm{err}_{\Pi}(h^*(\Pi))$. For a set $S$, we abuse notation and use $S$ to also denote the uniform distribution over the elements of $S$. We define $\mathbb{P}_{\Pi}(\cdot) := \mathbb{P}_{(x,y)\sim\Pi}(\cdot)$, $\mathbb{E}_{\Pi}(\cdot) := \mathbb{E}_{(x,y)\sim\Pi}(\cdot)$.

Given access to examples from a data distribution $D$ through an example oracle $\mathcal{U}$ and a labeling oracle $\mathcal{O}$, we aim to provide a classifier $\hat{h} \in \mathcal{H}$ such that with probability $\geq 1 - \delta$, $\mathrm{err}_D(\hat{h}) \leq \nu^*(D) + \epsilon$, for some target values of $\epsilon$ and $\delta$; this is achieved in an adaptive manner by making as few queries to the labelling oracle $\mathcal{O}$ as possible. When $\nu^*(D) = 0$, we are said to be in the *realizable case*; in the more general *agnostic* case, we make no assumptions on the labels, and thus $\nu^*(D)$ can be positive.

Previous approaches to agnostic active learning have frequently used the notion of *disagreements*. The disagreement between two hypotheses $h_1$ and $h_2$ with respect to a data distribution $\Pi$ is the fraction of examples according to $\Pi$ to which $h_1$ and $h_2$ assign different labels; formally: $\rho_{\Pi}(h_1, h_2) = \mathbb{P}_{(x,y)\sim\Pi}(h_1(x) \neq h_2(x))$. Observe that a data distribution $\Pi$ induces a pseudo-metric $\rho_{\Pi}$ on the elements of $\mathcal{H}$; this is called the disagreement metric. For any $r$ and any $h \in \mathcal{H}$, define $B_{\Pi}(h, r)$ to be the disagreement ball of radius $r$ around $h$ with respect to the data distribution $\Pi$. Formally: $B_{\Pi}(h, r) = \{h' \in \mathcal{H} : \rho_{\Pi}(h, h') \leq r\}$.

For notational simplicity, we assume that the hypothesis space is "dense" with repsect to the data distribution $D$, in the sense that $\forall r > 0$, $\sup_{h\in B_D(h^*(D),r)} \rho_D(h, h^*(D)) = r$. Our analysis will still apply without the denseness assumption, but will be significantly more messy. Finally, given a set of hypotheses $V \subseteq \mathcal{H}$, the *disagreement region* of $V$ is the set of all examples $x$ such that there exist two hypotheses $h_1, h_2 \in V$ for which $h_1(x) \neq h_2(x)$.

This paper establishes a connection between active learning and confidence-rated predictors with guaranteed error. A confidence-rated predictor is a prediction algorithm that is occasionally allowed to abstain from classification. We will consider such predictors in the transductive setting. Given a set $V$ of candidate hypotheses, an error guarantee $\eta$, and a set $U$ of unlabelled examples, a confidence-rated predictor $P$ either assigns a label or abstains from prediction on each unlabelled

$x \in U$. The labels are assigned with the guarantee that the expected disagreement[1] between the label assigned by $P$ and any $h \in V$ is $\leq \eta$. Specifically,

$$\text{for all } h \in V, \quad \mathbb{P}_{x \sim U}(h(x) \neq P(x), P(x) \neq 0) \leq \eta \tag{1}$$

This ensures that if some $h^* \in V$ is the true risk minimizer, then, the labels predicted by $P$ on $U$ do not differ very much from those predicted by $h^*$. The performance of a confidence-rated predictor which has a guarantee such as in Equation (1) is measured by its *coverage*, or the probability of non-abstention $\mathbb{P}_{x \sim U}(P(x) \neq 0)$; higher coverage implies better performance.

## 2.2 Main Algorithm

Our active learning algorithm proceeds in epochs, where the goal of epoch $k$ is to achieve excess generalization error $\epsilon_k = \epsilon 2^{k_0 - k + 1}$, by querying a fresh batch of labels. The algorithm maintains a candidate set $V_k$ that is guaranteed to contain the true risk minimizer.

The critical decision at each epoch is how to select a subset of unlabelled examples whose labels should be queried. We make this decision using a confidence-rated predictor $P$. At epoch $k$, we run $P$ with candidate hypothesis set $V = V_k$ and error guarantee $\eta = \epsilon_k/64$. Whenever $P$ abstains, we query the label of the example. The number of labels $m_k$ queried is adjusted so that it is enough to achieve excess generalization error $\epsilon_{k+1}$.

An outline is described in Algorithm 1; we next discuss each individual component in detail.

---

**Algorithm 1** Active Learning Algorithm: Outline

1: **Inputs:** Example oracle $\mathcal{U}$, Labelling oracle $\mathcal{O}$, hypothesis class $\mathcal{H}$ of VC dimension $d$, confidence-rated predictor $P$, target excess error $\epsilon$ and target confidence $\delta$.
2: Set $k_0 = \lceil \log 1/\epsilon \rceil$. Initialize candidate set $V_1 = \mathcal{H}$.
3: **for** $k = 1, 2, ..k_0$ **do**
4:     Set $\epsilon_k = \epsilon 2^{k_0 - k + 1}$, $\delta_k = \frac{\delta}{2(k_0 - k + 1)^2}$.
5:     Call $\mathcal{U}$ to generate a fresh unlabelled sample $U_k = \{z_{k,1}, ..., z_{k,n_k}\}$ of size $n_k = 192(\frac{512}{\epsilon_k})^2(d \ln 192(\frac{512}{\epsilon_k})^2 + \ln \frac{288}{\delta_k})$.
6:     Run confidence-rated predictor $P$ with inpuy $V = V_k$, $U = U_k$ and error guarantee $\eta = \epsilon_k/64$ to get abstention probabilities $\gamma_{k,1}, ..., \gamma_{k,n_k}$ on the examples in $U_k$. These probabilities induce a distribution $\Gamma_k$ on $U_k$. Let $\phi_k = \mathbb{P}_{x \sim U_k}(P(x) = 0) = \frac{1}{n_k} \sum_{i=1}^{n_k} \gamma_{k,i}$.
7:     **if** in the Realizable Case **then**
8:         Let $m_k = \frac{1536\phi_k}{\epsilon_k}(d \ln \frac{1536\phi_k}{\epsilon_k} + \ln \frac{48}{\delta_k})$. Draw $m_k$ i.i.d examples from $\Gamma_k$ and query $\mathcal{O}$ for labels of these examples to get a labelled data set $S_k$. Update $V_{k+1}$ using $S_k$: $V_{k+1} := \{h \in V_k : h(x) = y, \text{ for all } (x, y) \in S_k\}$.
9:     **else**
10:         In the non-realizable case, use Algorithm 2 with inputs hypothesis set $V_k$, distribution $\Gamma_k$, target excess error $\frac{\epsilon_k}{8\phi_k}$, target confidence $\frac{\delta_k}{2}$, and the labeling oracle $\mathcal{O}$ to get a new hypothesis set $V_{k+1}$.
11: **return** an arbitrary $\hat{h} \in V_{k_0+1}$.

---

**Candidate Sets.** At epoch $k$, we maintain a set $V_k$ of candidate hypotheses guaranteed to contain the true risk minimizer $h^*(D)$ (w.h.p). In the realizable case, we use a version space as our candidate set. The version space with respect to a set $S$ of labelled examples is the set of all $h \in \mathcal{H}$ such that $h(x_i) = y_i$ for all $(x_i, y_i) \in S$.

**Lemma 1.** *Suppose we run Algorithm 1 in the realizable case with inputs example oracle $\mathcal{U}$, labelling oracle $\mathcal{O}$, hypothesis class $\mathcal{H}$, confidence-rated predictor $P$, target excess error $\epsilon$ and target confidence $\delta$. Then, with probability 1, $h^*(D) \in V_k$, for all $k = 1, 2, \ldots, k_0 + 1$.*

In the non-realizable case, the version space is usually empty; we use instead a $(1 - \alpha)$-confidence set for the true risk minimizer. Given a set $S$ of $n$ labelled examples, let $C(S) \subseteq \mathcal{H}$ be a function of

$S$; $C(S)$ is said to be a $(1 - \alpha)$-confidence set for the true risk minimizer if for all data distributions $\Delta$ over $\mathcal{X} \times \mathcal{Y}$,

$$\mathbb{P}_{S \sim \Delta^n}[h^*(\Delta) \in C(S)] \geq 1 - \alpha,$$

Recall that $h^*(\Delta) = \operatorname{argmin}_{h \in \mathcal{H}} \operatorname{err}_\Delta(h)$. In the non-realizable case, our candidate sets are $(1 - \alpha)$-confidence sets for $h^*(D)$, for $\alpha = \delta$. The precise setting of $V_k$ is explained in Algorithm 2.

**Lemma 2.** *Suppose we run Algorithm 1 in the non-realizable case with inputs example oracle $\mathcal{U}$, labelling oracle $\mathcal{O}$, hypothesis class $\mathcal{H}$, confidence-rated predictor $P$, target excess error $\epsilon$ and target confidence $\delta$. Then with probability $1 - \delta$, $h^*(D) \in V_k$, for all $k = 1, 2, \ldots, k_0 + 1$.*

**Label Query.** We next discuss our label query procedure – which examples should we query labels for, and how many labels should we query at each epoch?

**Which Labels to Query?** Our goal is to query the labels of the most informative examples. To choose these examples while still maintaining consistency, we use a confidence-rated predictor $P$ with guaranteed error. The inputs to the predictor are our candidate hypothesis set $V_k$ which contains (w.h.p) the true risk minimizer, a fresh set $U_k$ of unlabelled examples, and an error guarantee $\eta = \epsilon_k/64$. For notation simplicity, assume the elements in $U_k$ are distinct. The output is a sequence of abstention probabilities $\{\gamma_{k,1}, \gamma_{k,2}, \ldots, \gamma_{k,n_k}\}$, for each example in $U_k$. It induces a distribution $\Gamma_k$ over $U_k$, from which we independently draw examples for label queries.

**How Many Labels to Query?** The goal of epoch $k$ is to achieve excess generalization error $\epsilon_k$. To achieve this, passive learning requires $\tilde{O}(d/\epsilon_k)$ labelled examples[2] in the realizable case, and $\tilde{O}(d(\nu^*(D) + \epsilon_k)/\epsilon_k^2)$ examples in the agnostic case. A key observation in this paper is that in order to achieve excess generalization error $\epsilon_k$ on $D$, it suffices to achieve a much larger excess generalization error $O(\epsilon_k/\phi_k)$ on the data distribution induced by $\Gamma_k$ and $D_{Y|X}$, where $\phi_k$ is the fraction of examples on which the confidence-rated predictor abstains.

In the realizable case, we achieve this by sampling $m_k = \frac{1536\phi_k}{\epsilon_k}(d \ln \frac{1536\phi_k}{\epsilon_k} + \ln \frac{48}{\delta_k})$ i.i.d examples from $\Gamma_k$, and querying their labels to get a labelled dataset $S_k$. Observe that as $\phi_k$ is the abstention probability of $P$ with guaranteed error $\leq \epsilon_k/64$, it is generally smaller than the measure of the disagreement region of the version space; this key fact results in improved label complexity over disagreement-based active learning. This sampling procedure has the following property:

**Lemma 3.** *Suppose we run Algorithm 1 in the realizable case with inputs example oracle $\mathcal{U}$, labelling oracle $\mathcal{O}$, hypothesis class $\mathcal{H}$, confidence-rated predictor $P$, target excess error $\epsilon$ and target confidence $\delta$. Then with probability $1 - \delta$, for all $k = 1, 2, \ldots, k_0 + 1$, and for all $h \in V_k$, $\operatorname{err}_D(h) \leq \epsilon_k$. In particular, the $\hat{h}$ returned at the end of the algorithm satisfies $\operatorname{err}_D(\hat{h}) \leq \epsilon$.*

The agnostic case has an added complication – in practice, the value of $\nu^*$ is not known ahead of time. Inspired by [24], we use a *doubling procedure*(stated in Algorithm 2) which adaptively finds the number $m_k$ of labelled examples to be queried and queries them. The following two lemmas illustrate its properties – that it is consistent, and that it does not use too many label queries.

**Lemma 4.** *Suppose we run Algorithm 2 with inputs hypothesis set $V$, example distribution $\Delta$, labelling oracle $\mathcal{O}$, target excess error $\tilde{\epsilon}$ and target confidence $\tilde{\delta}$. Let $\tilde{\Delta}$ be the joint distribution on $\mathcal{X} \times \mathcal{Y}$ induced by $\Delta$ and $D_{Y|X}$. Then there exists an event $\tilde{E}$, $\mathbb{P}(\tilde{E}) \geq 1 - \tilde{\delta}$, such that on $\tilde{E}$, (1) Algorithm 2 halts and (2) the set $V_{j_0}$ has the following properties:*

*(2.1) If for $h \in \mathcal{H}$, $\operatorname{err}_{\tilde{\Delta}}(h) - \operatorname{err}_{\tilde{\Delta}}(h^*(\tilde{\Delta})) \leq \tilde{\epsilon}/2$, then $h \in V_{j_0}$.*

*(2.2) On the other hand, if $h \in V_{j_0}$, then $\operatorname{err}_{\tilde{\Delta}}(h) - \operatorname{err}_{\tilde{\Delta}}(h^*(\tilde{\Delta})) \leq \tilde{\epsilon}$.*

When event $\tilde{E}$ happens, we say Algorithm 2 succeeds.

**Lemma 5.** *Suppose we run Algorithm 2 with inputs hypothesis set $V$, example distribution $\Delta$, labelling oracle $\mathcal{O}$, target excess error $\tilde{\epsilon}$ and target confidence $\tilde{\delta}$. There exists some absolute constant $c_1 > 0$, such that on the event that Algorithm 2 succeeds, $n_{j_0} \leq c_1((d \ln \frac{1}{\tilde{\epsilon}} + \ln \frac{1}{\tilde{\delta}})\frac{\nu^*(\tilde{\Delta}) + \tilde{\epsilon}}{\tilde{\epsilon}^2})$. Thus the total number of labels queried is $\sum_{j=1}^{j_0} n_j \leq 2n_{j_0} \leq 2c_1((d \ln \frac{1}{\tilde{\epsilon}} + \ln \frac{1}{\tilde{\delta}})\frac{\nu^*(\tilde{\Delta}) + \tilde{\epsilon}}{\tilde{\epsilon}^2})$.*

A naive approach (see Algorithm 4 in the Appendix) which uses an additive VC bound gives a sample complexity of $O((d\ln(1/\tilde{\epsilon}) + \ln(1/\tilde{\delta}))\tilde{\epsilon}^{-2})$; Algorithm 2 gives a better sample complexity.

The following lemma is a consequence of our label query procedure in the non-realizable case.

**Lemma 6.** *Suppose we run Algorithm 1 in the non-realizable case with inputs example oracle $\mathcal{U}$, labelling oracle $\mathcal{O}$, hypothesis class $\mathcal{H}$, confidence-rated predictor $P$, target excess error $\epsilon$ and target confidence $\delta$. Then with probability $1 - \delta$, for all $k = 1, 2, \ldots, k_0 + 1$, and for all $h \in V_k$, $err_D(h) \leq err_D(h^*(D)) + \epsilon_k$. In particular, the $\hat{h}$ returned at the end of the algorithm satisfies $err_D(\hat{h}) \leq err_D(h^*(D)) + \epsilon$.*

---

**Algorithm 2** An Adaptive Algorithm for Label Query Given Target Excess Error

1: **Inputs:** Hypothesis set $V$ of VC dimension $d$, Example distribution $\Delta$, Labeling oracle $\mathcal{O}$, target excess error $\tilde{\epsilon}$, target confidence $\tilde{\delta}$.
2: **for** $j = 1, 2, \ldots$ **do**
3:      Draw $n_j = 2^j$ i.i.d examples from $\Delta$; query their labels from $\mathcal{O}$ to get a labelled dataset $S_j$. Denote $\tilde{\delta}_j := \tilde{\delta}/(j(j+1))$.
4:      Train an ERM classifier $\hat{h}_j \in V$ over $S_j$.
5:      Define the set $V_j$ as follows:

$$V_j = \left\{ h \in V : \mathrm{err}_{S_j}(h) \leq \mathrm{err}_{S_j}(\hat{h}_j) + \frac{\tilde{\epsilon}}{2} + \sigma(n_j, \tilde{\delta}_j) + \sqrt{\sigma(n_j, \tilde{\delta}_j)\rho_{S_j}(h, \hat{h}_j)} \right\}$$

     Where $\sigma(n, \delta) := \frac{16}{n}(2d\ln\frac{2en}{d} + \ln\frac{24}{\delta})$.
6:      **if** $\sup_{h \in V_j}(\sigma(n_j, \tilde{\delta}_j) + \sqrt{\sigma(n_j, \tilde{\delta}_j)\rho_{S_j}(h, \hat{h}_j)}) \leq \frac{\tilde{\epsilon}}{6}$ **then**
7:          $j_0 = j$, **break**
8: **return** $V_{j_0}$.

---

## 2.3 Confidence-Rated Predictor

Our active learning algorithm uses a confidence-rated predictor with guaranteed error to make its label query decisions. In this section, we provide a novel confidence-rated predictor with guaranteed error. This predictor has optimal coverage in the realizable case, and may be of independent interest. The predictor $P$ receives as input a set $V \subseteq \mathcal{H}$ of hypotheses (which is likely to contain the true risk minimizer), an error guarantee $\eta$, and a set of $U$ of unlabelled examples. We consider a *soft prediction algorithm*; so, for each example in $U$, the predictor $P$ outputs three probabilities that add up to $1$ – the probability of predicting $1$, $-1$ and $0$. This output is subject to the constraint that the expected disagreement[3] between the $\pm 1$ labels assigned by $P$ and those assigned by any $h \in V$ is at most $\eta$, and the goal is to maximize the coverage, or the expected fraction of non-abstentions.

Our key insight is that this problem can be written as a linear program, which is described in Algorithm 3. There are three variables, $\xi_i$, $\zeta_i$ and $\gamma_i$, for each unlabelled $z_i \in U$; there are the probabilities with which we predict $1$, $-1$ and $0$ on $z_i$ respectively. Constraint (2) ensures that the expected disagreement between the label predicted and any $h \in V$ is no more than $\eta$, while the LP objective maximizes the coverage under these constraints. Observe that the LP is always feasible. Although the LP has infinitely many constraints, the number of constraints in Equation (2) distinguishable by $U_k$ is at most $(em/d)^d$, where $d$ is the VC dimension of the hypothesis class $\mathcal{H}$.

The performance of a confidence-rated predictor is measured by its error and coverage. The error of a confidence-rated predictor is the probability with which it predicts the wrong label on an example, while the coverage is its probability of non-abstention. We can show the following guarantee on the performance of the predictor in Algorithm 3.

**Theorem 1.** *In the realizable case, if the hypothesis set $V$ is the version space with respect to a training set, then $\mathbb{P}_{x \sim U}(P(x) \neq h^*(x), P(x) \neq 0) \leq \eta$. In the non-realizable case, if the hypothesis set $V$ is an $(1 - \alpha)$-confidence set for the true risk minimizer $h^*$, then, w.p $\geq 1 - \alpha$, $\mathbb{P}_{x \sim U}(P(x) \neq y, P(x) \neq 0) \leq \mathbb{P}_{x \sim U}(h^*(x) \neq y) + \eta$.*

**Algorithm 3** Confidence-rated Predictor
___

1: **Inputs:** hypothesis set $V$, unlabelled data $U = \{z_1, \dots, z_m\}$, error bound $\eta$.
2: Solve the linear program:

$$\min \sum_{i=1}^{m} \gamma_i$$

$$\text{subject to:} \quad \forall i, \ \xi_i + \zeta_i + \gamma_i = 1$$

$$\forall h \in V, \quad \sum_{i:h(z_i)=1} \zeta_i + \sum_{i:h(z_i)=-1} \xi_i \leq \eta m \qquad (2)$$

$$\forall i, \ \xi_i, \zeta_i, \gamma_i \geq 0$$

3: For each $z_i \in U$, output probabilities for predicting $1, -1$ and $0$: $\xi_i, \zeta_i$, and $\gamma_i$.
___

In the realizable case, we can also show that our confidence rated predictor has optimal coverage. Observe that we cannot directly show optimality in the non-realizable case, as the performance depends on the exact choice of the $(1 - \alpha)$-confidence set.

**Theorem 2.** *In the realizable case, suppose that the hypothesis set $V$ is the version space with respect to a training set. If $P'$ is any confidence rated predictor with error guarantee $\eta$, and if $P$ is the predictor in Algorithm 3, then, the coverage of $P$ is at least much as the coverage of $P'$.*

## 3    Performance Guarantees

An essential property of any active learning algorithm is consistency – that it converges to the true risk minimizer given enough labelled examples. We observe that our algorithm is consistent provided we use *any* confidence-rated predictor $P$ with guaranteed error as a subroutine. The consistency of our algorithm is a consequence of Lemmas 3 and 6 and is shown in Theorem 3.

**Theorem 3** (Consistency). *Suppose we run Algorithm 1 with inputs example oracle $\mathcal{U}$, labelling oracle $\mathcal{O}$, hypothesis class $\mathcal{H}$, confidence-rated predictor $P$, target excess error $\epsilon$ and target confidence $\delta$. Then with probability $1 - \delta$, the classifier $\hat{h}$ returned by Algorithm 1 satisfies $err_D(\hat{h}) - err_D(h^*(D)) \leq \epsilon$.*

We now establish a label complexity bound for our algorithm; however, this label complexity bound applies only if we use the predictor described in Algorithm 3 as a subroutine.

For any hypothesis set $V$, data distribution $D$, and $\eta$, define $\boldsymbol{\Phi}_D(V, \eta)$ to be the minimum abstention probability of a confidence-rated predictor which guarantees that the disagreement between its predicted labels and any $h \in V$ under $D_{\mathcal{X}}$ is at most $\eta$.

Formally, $\boldsymbol{\Phi}_D(V, \eta) = \min\{\mathbb{E}_D \gamma(x) : \mathbb{E}_D[I(h(x) = +1)\zeta(x) + I(h(x) = -1)\xi(x)] \leq \eta$ for all $h \in V, \gamma(x) + \xi(x) + \zeta(x) \equiv 1, \gamma(x), \xi(x), \zeta(x) \geq 0\}$. Define $\phi(r, \eta) := \boldsymbol{\Phi}_D(B_D(h^*, r), \eta)$. The label complexity of our active learning algorithm can be stated as follows.

**Theorem 4** (Label Complexity). *Suppose we run Algorithm 1 with inputs example oracle $\mathcal{U}$, labelling oracle $\mathcal{O}$, hypothesis class $\mathcal{H}$, confidence-rated predictor $P$ of Algorithm 3, target excess error $\epsilon$ and target confidence $\delta$. Then there exist constants $c_3, c_4 > 0$ such that with probability $1 - \delta$:*
*(1) In the realizable case, the total number of labels queried by Algorithm 1 is at most:*

$$c_3 \sum_{k=1}^{\lceil \log \frac{1}{\epsilon} \rceil} \left( d \ln \frac{\phi(\epsilon_k, \epsilon_k/256)}{\epsilon_k} + \ln\left( \frac{\lceil \log(1/\epsilon) \rceil - k + 1}{\delta} \right) \right) \frac{\phi(\epsilon_k, \epsilon_k/256)}{\epsilon_k}$$

*(2) In the agnostic case, the total number of labels queried by Algorithm 1 is at most:*

$$c_4 \sum_{k=1}^{\lceil \log \frac{1}{\epsilon} \rceil} \left( d \ln \frac{\phi(2\nu^*(D) + \epsilon_k, \epsilon_k/256)}{\epsilon_k} + \ln\left( \frac{\lceil \log(1/\epsilon) \rceil - k + 1}{\delta} \right) \right) \frac{\phi(2\nu^*(D) + \epsilon_k, \epsilon_k/256)}{\epsilon_k} \left( 1 + \frac{\nu^*(D)}{\epsilon_k} \right)$$

**Comparison.** The label complexity of disagreement-based active learning is characterized in terms of the *disagreement coefficient*. Given a radius $r$, the disagreement coefficent $\theta(r)$ is defined as:

$$\theta(r) = \sup_{r' \geq r} \frac{\mathbb{P}(\text{DIS}(B_D(h^*, r')))}{r'},$$

where for any $V \subseteq \mathcal{H}$, $\text{DIS}(V)$ is the disagreement region of $V$. As $\mathbb{P}(\text{DIS}(B_D(h^*, r))) = \phi(r, 0)$ [13], in our notation, $\theta(r) = \sup_{r' \geq r} \frac{\phi(r', 0)}{r'}$.

In the realizable case, the best known bound for label complexity of disagreement-based active learning is $\tilde{O}(\theta(\epsilon) \cdot \ln(1/\epsilon) \cdot (d \ln \theta(\epsilon) + \ln \ln(1/\epsilon)))$ [20][4]. Our label complexity bound may be simplified to:

$$\tilde{O}\left( \ln \frac{1}{\epsilon} \cdot \sup_{k \leq \lceil \log(1/\epsilon) \rceil} \frac{\phi(\epsilon_k, \epsilon_k/256)}{\epsilon_k} \cdot \left( d \ln \left( \sup_{k \leq \lceil \log(1/\epsilon) \rceil} \frac{\phi(\epsilon_k, \epsilon_k/256)}{\epsilon_k} \right) + \ln \ln \frac{1}{\epsilon} \right) \right),$$

which is essentially the bound of [20] with $\theta(\epsilon)$ replaced by $\sup_{k \leq \lceil \log(1/\epsilon) \rceil} \frac{\phi(\epsilon_k, \epsilon_k/256)}{\epsilon_k}$. As enforcing a lower error guarantee requires more abstention, $\phi(r, \eta)$ is a decreasing function of $\eta$; as a result,

$$\sup_{k \leq \lceil \log(1/\epsilon) \rceil} \frac{\phi(\epsilon_k, \epsilon_k/256)}{\epsilon_k} \leq \theta(\epsilon),$$

and our label complexity bound is better.

In the agnostic case, [12] provides a label complexity bound of $\tilde{O}(\theta(2\nu^*(D)+\epsilon) \cdot (d\frac{\nu^*(D)^2}{\epsilon^2} \ln(1/\epsilon) + d \ln^2(1/\epsilon)))$ for disagreement-based active-learning. In contrast, by Proposition 1 our label complexity is at most:

$$\tilde{O}\left( \sup_{k \leq \lceil \log(1/\epsilon) \rceil} \frac{\phi(2\nu^*(D) + \epsilon_k, \epsilon_k/256)}{2\nu^*(D) + \epsilon_k} \cdot \left( d\frac{\nu^*(D)^2}{\epsilon^2} \ln(1/\epsilon) + d \ln^2(1/\epsilon) \right) \right)$$

Again, this is essentially the bound of [12] with $\theta(2\nu^*(D) + \epsilon)$ replaced by the smaller quantity

$$\sup_{k \leq \lceil \log(1/\epsilon) \rceil} \frac{\phi(2\nu^*(D) + \epsilon_k, \epsilon_k/256)}{2\nu^*(D) + \epsilon_k},$$

[20] has provided a more refined analysis of disagreement-based active learning that gives a label complexity of $\tilde{O}(\theta(\nu^*(D) + \epsilon)(\frac{\nu^*(D)^2}{\epsilon^2} + \ln\frac{1}{\epsilon})(d \ln \theta(\nu^*(D) + \epsilon) + \ln \ln \frac{1}{\epsilon}))$; observe that their dependence is still on $\theta(\nu^*(D) + \epsilon)$. We leave a more refined label complexity analysis of our algorithm for future work.

An important sub-case of learning from noisy data is learning under the Tsybakov noise conditions [30]. We defer the discussion into the Appendix.

## 3.1 Case Study: Linear Classification under the Log-concave Distribution

We now consider learning linear classifiers with respect to log-concave data distribution on $\mathbf{R}^d$. In this case, for any $r$, the disagreement coefficient $\theta(r) \leq O(\sqrt{d} \ln(1/r))$ [4]; however, for any $\eta > 0$, $\frac{\phi(r,\eta)}{r} \leq O(\ln(r/\eta))$ (see Lemma 14 in the Appendix), which is much smaller so long as $\eta/r$ is not too small. This leads to the following label complexity bounds.

**Corollary 1.** *Suppose $D_{\mathcal{X}}$ is isotropic and log-concave on $\mathbf{R}^d$, and $\mathcal{H}$ is the set of homogeneous linear classifiers on $\mathbf{R}^d$. Then Algorithm 1 with inputs example oracle $\mathcal{U}$, labelling oracle $\mathcal{O}$, hypothesis class $\mathcal{H}$, confidence-rated predictor $P$ of Algorithm 3, target excess error $\epsilon$ and target confidence $\delta$ satisfies the following properties. With probability $1 - \delta$:*
*(1) In the realizable case, there exists some absolute constant $c_8 > 0$ such that the total number of labels queried is at most $c_8 \ln \frac{1}{\epsilon}(d + \ln \ln \frac{1}{\epsilon} + \ln \frac{1}{\delta})$.*

*(2) In the agnostic case, there exists some absolute constant $c_9 > 0$ such that the total number of labels queried is at most $c_9\big(\frac{\nu^*(D)^2}{\epsilon^2} + \ln\frac{1}{\epsilon}\big)\ln\frac{\epsilon+\nu^*(D)}{\epsilon}\big(d\ln\frac{\epsilon+\nu^*(D)}{\epsilon} + \ln\frac{1}{\delta}\big) + \ln\frac{1}{\epsilon}\ln\frac{\epsilon+\nu^*(D)}{\epsilon}\ln\ln\frac{1}{\epsilon}$.*
*(3) If $(C_0,\kappa)$-Tsybakov Noise condition holds for $D$ with respect to $\mathcal{H}$, then there exists some constant $c_{10} > 0$ (that depends on $C_0, \kappa$) such that the total number of labels queried is at most $c_{10}\epsilon^{\frac{2}{\kappa}-2}\ln\frac{1}{\epsilon}(d\ln\frac{1}{\epsilon} + \ln\frac{1}{\delta})$.*

In the realizable case, our bound matches [4]. For disagreement-based algorithms, the bound is $\tilde{O}(d^{\frac{3}{2}}\ln^2\frac{1}{\epsilon}(\ln d + \ln\ln\frac{1}{\epsilon}))$, which is worse by a factor of $O(\sqrt{d}\ln(1/\epsilon))$. [4] does not address the fully agnostic case directly; however, if $\nu^*(D)$ is known a-priori, then their algorithm can achieve roughly the same label complexity as ours.

For the Tsybakov Noise Condition with $\kappa > 1$, [3, 4] provides a label complexity bound for $\tilde{O}(\epsilon^{\frac{2}{\kappa}-2}\ln^2\frac{1}{\epsilon}(d + \ln\ln\frac{1}{\epsilon}))$ with an algorithm that has a-priori knowledge of $C_0$ and $\kappa$. We get a slightly better bound. On the other hand, a disagreement based algorithm [20] gives a label complexity of $\tilde{O}(d^{\frac{3}{2}}\ln^2\frac{1}{\epsilon}\epsilon^{\frac{2}{\kappa}-2}(\ln d + \ln\ln\frac{1}{\epsilon}))$. Again our bound is better by factor of $\Omega(\sqrt{d})$ over disagreement-based algorithms. For $\kappa = 1$, we can tighten our label complexity to get a $\tilde{O}(\ln\frac{1}{\epsilon}(d + \ln\ln\frac{1}{\epsilon} + \ln\frac{1}{\delta}))$ bound, which again matches [4], and is better than the ones provided by disagreement-based algorithm – $\tilde{O}(d^{\frac{3}{2}}\ln^2\frac{1}{\epsilon}(\ln d + \ln\ln\frac{1}{\epsilon}))$ [20].

## 4 Related Work

Active learning has seen a lot of progress over the past two decades, motivated by vast amounts of unlabelled data and the high cost of annotation [28, 10, 20]. According to [10], the two main threads of research are exploitation of cluster structure [31, 11], and efficient search in hypothesis space, which is the setting of our work. We are given a hypothesis class $\mathcal{H}$, and the goal is to find an $h \in \mathcal{H}$ that achieves a target excess generalization error, while minimizing the number of label queries.

Three main approaches have been studied in this setting. The first and most natural one is generalized binary search [17, 8, 9, 27], which was analyzed in the realizable case by [9] and in various limited noise settings by [23, 27, 26]. While this approach has the advantage of low label complexity, it is generally inconsistent in the fully agnostic setting [11]. The second approach, disagreement-based active learning, is consistent in the agnostic PAC model. [7] provides the first disagreement-based algorithm for the realizable case. [2] provides an agnostic disagreement-based algorithm, which is analyzed in [18] using the notion of disagreement coefficient. [12] reduces disagreement-based active learning to passive learning; [5] and [6] further extend this work to provide practical and efficient implementations. [19, 24] give algorithms that are adaptive to the Tsybakov Noise condition. The third line of work [3, 4, 1], achieves a better label complexity than disagreement-based active learning for linear classifiers on the uniform distribution over unit sphere and logconcave distributions. However, a limitation is that their algorithm applies only to these specific settings, and it is not apparent how to apply it generally.

Research on confidence-rated prediction has been mostly focused on empirical work, with relatively less theoretical development. Theoretical work on this topic includes KWIK learning [25], conformal prediction [29] and the weighted majority algorithm of [16]. The closest to our work is the recent learning-theoretic treatment by [13, 14]. [13] addresses confidence-rated prediction with guaranteed error in the realizable case, and provides a predictor that abstains in the disagreement region of the version space. This predictor achieves zero error, and coverage equal to the measure of the agreement region. [14] shows how to extend this algorithm to the non-realizable case and obtain zero error with respect to the best hypothesis in $\mathcal{H}$. Note that the predictors in [13, 14] generally achieve less coverage than ours for the same error guarantee; in fact, if we plug them into our Algorithm 1, then we recover the label complexity bounds of disagreement-based algorithms [12, 19, 24].

A formal connection between disagreement-based active learning in realizable case and perfect confidence-rated prediction (with a zero error guarantee) was established by [15]. Our work can be seen as a step towards bridging these two areas, by demonstrating that active learning can be further reduced to imperfect confidence-rated prediction, with potentially higher label savings.

**Acknowledgements.** We thank NSF under IIS-1162581 for research support. We thank Sanjoy Dasgupta and Yoav Freund for helpful discussions. CZ would like to thank Liwei Wang for introducing the problem of selective classification to him.

## Footnotes

[1] where the expectation is with respect to the random choices made by $P$

[2]$\tilde{O}(\cdot)$ hides logarithmic factors

[3]where the expectation is taken over the random choices made by $P$

[4]Here the $\tilde{O}(\cdot)$ notation hides factors logarithmic in $1/\delta$

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
