[Supplementary Material · nips14_supp.pdf]

## A  Tsybakov Noise Conditions

An important sub-case of learning from noisy data is learning under the Tsybakov noise conditions [30].

**Definition 1.** *(Tsybakov Noise Condition) Let $\kappa \geq 1$. A labelled data distribution $D$ over $\mathcal{X} \times \mathcal{Y}$ satisfies $(C_0, \kappa)$-Tsybakov Noise Condition with respect to a hypothesis class $\mathcal{H}$ for some constant $C_0 > 0$, if for all $h \in \mathcal{H}$, $\rho_D(h, h^*(D)) \leq C_0(err_D(h) - err_D(h^*(D)))^{\frac{1}{\kappa}}$.*

The following theorem shows the performance guarantees achieved by Algorithm 1 under the Tsybakov noise conditions.

**Theorem 5.** *Suppose $(C_0, \kappa)$-Tsybakov Noise Condition holds for $D$ with respect to $\mathcal{H}$. Then Algorithm 1 with inputs example oracle $\mathcal{U}$, labelling oracle $\mathcal{O}$, hypothesis class $\mathcal{H}$, confidence-rated predictor $P$ of Algorithm 3, target excess error $\epsilon$ and target confidence $\delta$ satisfies the following properties. There exists a constant $c_5 > 0$ such that with probability $1 - \delta$, the total number of labels queried by Algorithm 1 is at most:*

$$c_5 \sum_{k=1}^{\lceil \log \frac{1}{\epsilon} \rceil} (d \ln(\phi(C_0 \epsilon_k^{\frac{1}{\kappa}}, \epsilon_k/256) \epsilon_k^{\frac{1}{\kappa}-2}) + \ln(\frac{\lceil \log \frac{1}{\epsilon} \rceil - k + 1}{\delta})) \phi(C_0 \epsilon_k^{\frac{1}{\kappa}}, \epsilon_k/256) \epsilon_k^{\frac{1}{\kappa}-2}$$

**Comparison.** [20] provides a label complexity bound of $\tilde{O}(\theta(C_0 \epsilon^{\frac{1}{\kappa}}) \epsilon^{\frac{2}{\kappa}-2} \ln \frac{1}{\epsilon} (d \ln \theta(C_0 \epsilon^{\frac{1}{\kappa}}) + \ln \ln \frac{1}{\epsilon}))$ for disagreement-based active learning. For $\kappa > 1$, by Proposition 2, our label complexity is at most:

$$\tilde{O}\left( \sup_{k \leq \lceil \log(1/\epsilon) \rceil} \frac{\phi(C_0 \epsilon_k^{1/\kappa}, \epsilon_k/256)}{\epsilon_k^{1/\kappa}} \cdot \epsilon_k^{2/\kappa-2} \cdot d \ln(1/\epsilon) \right),$$

For $\kappa = 1$, our label complexity is at most

$$\tilde{O}\left( \ln \frac{1}{\epsilon} \cdot \sup_{k \leq \lceil \log(1/\epsilon) \rceil} \frac{\phi(C_0 \epsilon_k, \epsilon_k/256)}{\epsilon_k} \cdot \left( d \ln(\sup_{k \leq \lceil \log(1/\epsilon) \rceil} \frac{\phi(C_0 \epsilon_k, \epsilon_k/256)}{\epsilon_k}) + \ln \ln \frac{1}{\epsilon} \right) \right).$$

In both cases, our bounds are better, as $\sup_{k \leq \lceil \log(1/\epsilon) \rceil} \cdot \frac{\phi(C_0 \epsilon_k^{1/\kappa}, \epsilon_k/256)}{C_0 \epsilon_k^{1/\kappa}} \leq \theta(C_0 \epsilon^{1/\kappa})$. In further work, [21] provides a refined analysis with a bound of $\tilde{O}(\theta(C_0 \epsilon^{\frac{1}{\kappa}}) \epsilon^{\frac{2}{\kappa}-2} d \ln \theta(C_0 \epsilon^{\frac{1}{\kappa}}))$; however, this work is not directly comparable to ours, as they need prior knowledge of $C_0$ and $\kappa$.

## B  Additional Notation and Concentration Lemmas

We begin with some additional notation that will be used in the subsequent proofs. Recall that we define:

$$\sigma(n, \delta) = \frac{16}{n}(2d \ln \frac{2en}{d} + \ln \frac{24}{\delta}), \tag{3}$$

where $d$ is the VC dimension of the hypothesis class $\mathcal{H}$.

The following lemma is an immediate corollary of the multiplicative VC bound; we pick the version of the multiplicative VC bound due to [22].

**Lemma 7.** *Pick any $n \geq 1$, $\delta \in (0, 1)$. Let $S_n$ be a set of $n$ iid copies of $(X, Y)$ drawn from a distribution $D$ over labelled examples. Then, the following hold with probability at least $1 - \delta$ over the choice of $S_n$:*
*(1) For all $h \in \mathcal{H}$,*

$$|err_D(h) - err_{S_n}(h)| \leq \min(\sigma(n, \delta) + \sqrt{\sigma(n, \delta)err_D(h)}, \sigma(n, \delta) + \sqrt{\sigma(n, \delta)err_{S_n}(h)}) \tag{4}$$

*In particular, all classifiers $h$ in $\mathcal{H}$ consistent with $S_n$ satisfies*

$$err_D(h) \leq \sigma(n, \delta) \tag{5}$$

*(2) For all $h, h'$ in $\mathcal{H}$,*

$$|(err_D(h)-err_D(h'))-(err_{S_n}(h)-err_{S_n}(h'))| \leq \sigma(n,\delta)+\min(\sqrt{\sigma(n,\delta)\rho_D(h,h')}, \sqrt{\sigma(n,\delta)\rho_{S_n}(h,h')})$$
$$(6)$$

$$|\rho_D(h,h') - \rho_{S_n}(h,h')| \leq \sigma(n,\delta) + \min(\sqrt{\sigma(n,\delta)\rho_D(h,h')}, \sqrt{\sigma(n,\delta)\rho_{S_n}(h,h')}) \quad (7)$$

*Where $\sigma(n,\delta)$ is defined in Equation (3).*

We occasionally use the following (weaker) version of Lemma 7.

**Lemma 8.** *Pick any $n \geq 1$, $\delta \in (0,1)$. Let $S_n$ be a set of $n$ iid copies of $(X,Y)$. The following holds with probability at least $1 - \delta$: (1) For all $h \in \mathcal{H}$,*

$$|err_D(h) - err_{S_n}(h)| \leq \sqrt{4\sigma(n,\delta)} \quad (8)$$

*(2) For all $h, h'$ in $\mathcal{H}$,*

$$|(err_D(h) - err_D(h')) - (err_{S_n}(h) - err_{S_n}(h'))| \leq \sqrt{4\sigma(n,\delta)} \quad (9)$$

$$|\rho_D(h,h') - \rho_{S_n}(h,h')| \leq \sqrt{4\sigma(n,\delta)} \quad (10)$$

*Where $\sigma(n,\delta)$ is defined in Equation (3).*

For an unlabelled sample $U_k$, we use $\tilde{U}_k$ to denote the joint distribution over $\mathcal{X} \times \mathcal{Y}$ induced by uniform distribution over $U_k$ and $D_{Y|X}$. We have:

**Lemma 9.** *If the size of $n_k$ of the unlabelled dataset $U_k$ is at least $192(\frac{512}{\epsilon_k})^2(d\ln 192(\frac{512}{\epsilon_k})^2 + \ln\frac{288}{\delta_k})$, then with probability $1 - \delta_k/4$, the following conditions hold for all $h, h' \in V_k$:*

$$|err_D(h) - err_{\tilde{U}_k}(h)| \leq \frac{\epsilon_k}{64} \quad (11)$$

$$|(err_D(h) - err_D(h')) - (err_{\tilde{U}_k}(h) - err_{\tilde{U}_k}(h'))| \leq \frac{\epsilon_k}{32} \quad (12)$$

$$|\rho_D(h,h') - \rho_{\tilde{U}_k}(h,h')| \leq \frac{\epsilon_k}{64} \quad (13)$$

**Lemma 10.** *If the size of $n_k$ of the unlabelled dataset $U_k$ is at least $192(\frac{512}{\epsilon_k})^2(d\ln 192(\frac{512}{\epsilon_k})^2 + \ln\frac{288}{\delta_k})$, then with probability $1 - \delta_k/4$, the following hold:*
*(1) The outputs $\{(\xi_{k,i}, \zeta_{k,i}, \gamma_{k,i})\}_{i=1}^{n_k}$ of any confidence-rated predictor with inputs hypothesis set $V_k$, unlabelled data $U_k$, and error bound $\epsilon_k/64$ satisfy:*

$$\frac{1}{n_k}\sum_{i=1}^{n_k}[I(h(x_i) \neq h'(x_i))(1 - \gamma_{k,i})] \leq \frac{\epsilon_k}{32}; \quad (14)$$

*(2) The outputs $\{(\xi_{k,i}, \zeta_{k,i}, \gamma_{k,i})\}_{i=1}^{n_k}$ of the confidence-rated predictor of Algortihm 3 with inputs hypothesis set $V_k$, unlabelled data $U_k$, and error bound $\epsilon_k/64$ satisfy:*

$$\phi_k \leq \mathbf{\Phi}_D(V_k, \frac{\epsilon_k}{128}) + \frac{\epsilon_k}{256} \quad (15)$$

We use $\tilde{\Gamma}_k$ to denote the joint distribution over $\mathcal{X} \times \mathcal{Y}$ induced by $\Gamma_k$ and $D_{Y|X}$. Denote $\gamma_k(x) : \mathcal{X} \to [0,1]$, where $\gamma_k(x_i) = \gamma_{k,i}$, and 0 elsewhere. Clearly, $\Gamma_k(\{x\}) = \frac{\gamma_k(x)}{n_k\phi_k}$ and $\tilde{\Gamma}_k(\{(x,y)\}) = \frac{\tilde{U}_k(\{(x,y)\})\gamma_k(x)}{\phi_k}$. Also, Equations (14) and (15) of Lemma 10 can be restated as

$$\forall h, h' \in V_k, \mathbb{E}_{\tilde{U}_k}[(1 - \gamma_k(x))I(h(x) \neq h'(x))] \leq \frac{\epsilon_k}{32}$$

$$\mathbb{E}_{\tilde{U}_k}[\gamma_k(x)] = \phi_k \leq \mathbf{\Phi}_D(V_k, \frac{\epsilon_k}{128}) + \frac{\epsilon_k}{256}$$

In the realizable case, define event

$$E_r = \{\text{For all } k = 1, 2, \ldots, k_0: \text{Equations (11), (12), (13), (14), (15) hold for } \tilde{U}_k$$
$$\text{and all classifiers consistent with } S_k \text{ have error at most } \frac{\epsilon_k}{8\phi_k} \text{ with respect to } \tilde{\Gamma}_k \}.$$

**Fact 1.** $\mathbb{P}(E_r) \geq 1 - \delta$.

*Proof.* By Equation (5) of Lemma 7, with probability $1 - \delta_k/2$, if $h \in V_k$ is consistent with $S_k$, then
$$\text{err}_{\tilde{\Gamma}_k}(h) \leq \sigma(m_k, \delta_k/2)$$
Because $m_k = \frac{1536\phi_k}{\epsilon_k}(d\ln\frac{1536\phi_k}{\epsilon_k} + \ln\frac{48}{\delta_k})$, we have $\text{err}_{\tilde{\Gamma}_k}(h) \leq \epsilon_k/8\phi_k$. The fact follows from combining the fact above with Lemma 9 and Lemma 10, and the union bound. $\square$

In the non-realizable case, define event

$E_a = \{$For all $k = 1, 2, \ldots, k_0$: Equations (11), (12), (13), (14), (15) hold for $\tilde{U}_k$,

and Algorithm 2 succeeds with inputs hypothesis set $V = V_k$, example distribution $\Delta = \Gamma_k$,

labelling oracle $\mathcal{O}$, target excess error $\tilde{\epsilon} = \dfrac{\epsilon_k}{8\phi_k}$ and target confidence $\tilde{\delta} = \dfrac{\delta_k}{2}\}$.

**Fact 2.** $\mathbb{P}(E_a) \geq 1 - \delta$.

*Proof.* This is an immediate consequence of Lemma 9, Lemma 10, Lemma 4 and union bound. $\square$

Recall that we assume the hypothesis space is "dense", in the sense that $\forall r > 0$, $\sup_{h \in B_D(h^*(D),r)} \rho(h, h^*(D)) = r$. We will call this the "denseness assumption".

## C   Proofs related to the properties of Algorithm 2

We first establish some properties of Algorithm 2. The inputs to Algorithm 2 are a set $V$ of hypotheses of VC dimension $d$, an example distribution $\Delta$, a labeling oracle $\mathcal{O}$, a target excess error $\tilde{\epsilon}$ and a target confidence $\tilde{\delta}$.

We define the event

$\tilde{E} = \{$For all $j = 1, 2, \ldots$: Equations (4)-(7) hold for sample $S_j$ with $n = n_j$ and $\delta = \tilde{\delta}_j \}$

By union bound, $\mathbb{P}(\tilde{E}) \geq 1 - \sum_j \tilde{\delta}_j \geq 1 - \tilde{\delta}$.

*Proof.* (of Lemma 4) Assume $\tilde{E}$ happens. For the proof of (1), define $j_{max}$ as the smallest integer $j$ such that $\sigma(n_j, \tilde{\delta}_j) \leq \tilde{\epsilon}^2/144$. Since $n_{j_{max}}$ is a power of 2,

$$n_{j_{max}} \leq 2\min\{n = 1, 2, \ldots : \frac{16(2d\ln\frac{2en}{d} + \ln\frac{24\log n(\log n+1)}{\delta})}{n} \leq \frac{\tilde{\epsilon}^2}{144}\}$$

Thus, $n_{j_{max}} \leq 384\frac{144}{\tilde{\epsilon}^2}(d\ln 192\frac{144}{\tilde{\epsilon}^2} + \ln\frac{24}{\delta})$. Then in round $j_{max}$, the stopping criterion (6) of Algorithm 2 is satisified; thus, Algorithm 2 halts with $j_0 \leq j_{max}$.

To prove (2.1), we observe that as $h^*(\tilde{\Delta})$ is the risk minimizer in $V$, if $h$ satisfies $\text{err}_{\tilde{\Delta}}(h) - \text{err}_{\tilde{\Delta}}(h^*(\tilde{\Delta})) \leq \frac{\tilde{\epsilon}}{2}$, then $\text{err}_{\tilde{\Delta}}(h) - \text{err}_{\tilde{\Delta}}(\hat{h}_{j_0}) \leq \frac{\tilde{\epsilon}}{2}$. By Equation (6) of Lemma 7,

$$\begin{aligned}
(\text{err}_{S_{j_0}}(h) - \text{err}_{S_{j_0}}(\hat{h}_{j_0})) &\leq (\text{err}_{\tilde{\Delta}}(h) - \text{err}_{\tilde{\Delta}}(\hat{h}_{j_0})) + \sigma(n_{j_0}, \tilde{\delta}_{j_0}) + \sqrt{\sigma(n_{j_0}, \tilde{\delta}_{j_0})\rho_{S_{j_0}}(h, \hat{h}_{j_0})} \\
&\leq \frac{\tilde{\epsilon}}{2} + \sigma(n_{j_0}, \tilde{\delta}_{j_0}) + \sqrt{\sigma(n_{j_0}, \tilde{\delta}_{j_0})\rho_{S_{j_0}}(h, \hat{h}_{j_0})}
\end{aligned}$$

Hence $h \in V_{j_0}$.

For the proof of (2.2), note first that by (2.1), in particular, $h^*(\tilde{\Delta}) \in V_{j_0}$. Hence by Equation (6) of Lemma 7, and the stopping criterion Equation (6),

$$(\text{err}_{\tilde{\Delta}}(\hat{h}_{j_0}) - \text{err}_{\tilde{\Delta}}(h^*(\tilde{\Delta}))) - (\text{err}_{S_{j_0}}(\hat{h}_{j_0}) - \text{err}_{S_{j_0}}(h^*(\tilde{\Delta}))) \leq \sigma(n_{j_0}, \tilde{\delta}_{j_0}) + \sqrt{\sigma(n_{j_0}, \tilde{\delta}_{j_0})\rho_{S_{j_0}}(\hat{h}_{j_0}, h^*(\tilde{\Delta}))} \leq \frac{\tilde{\epsilon}}{6}$$

Thus,
$$\operatorname{err}_{\tilde{\Delta}}(\hat{h}_{j_0}) - \operatorname{err}_{\tilde{\Delta}}(h^*(\tilde{\Delta})) \leq \frac{\tilde{\epsilon}}{6} \tag{16}$$

On the other hand, if $h \in V_{j_0}$, then

$$(\operatorname{err}_{\tilde{\Delta}}(h) - \operatorname{err}_{\tilde{\Delta}}(\hat{h}_{j_0})) - (\operatorname{err}_{S_{j_0}}(h) - \operatorname{err}_{S_{j_0}}(\hat{h}_{j_0})) \leq \sigma(n_{j_0}, \tilde{\delta}_{j_0}) + \sqrt{\sigma(n_{j_0}, \tilde{\delta}_{j_0})\rho_{S_{j_0}}(h, \hat{h}_{j_0})} \leq \frac{\tilde{\epsilon}}{6}$$

By definition of $V_{j_0}$,

$$(\operatorname{err}_{S_{j_0}}(h) - \operatorname{err}_{S_{j_0}}(\hat{h}_{j_0})) \leq \sigma(n_{j_0}, \tilde{\delta}_{j_0}) + \sqrt{\sigma(n_{j_0}, \tilde{\delta}_{j_0})\rho_{S_{j_0}}(h, \hat{h}_{j_0})} + \frac{\tilde{\epsilon}}{2} \leq \frac{2\tilde{\epsilon}}{3}$$

Hence,

$$\operatorname{err}_{\tilde{\Delta}}(h) - \operatorname{err}_{\tilde{\Delta}}(\hat{h}_{j_0}) \leq \frac{5\tilde{\epsilon}}{6} \tag{17}$$

Combining Equations (16) and (17), we have

$$\operatorname{err}_{\tilde{\Delta}}(h) - \operatorname{err}_{\tilde{\Delta}}(h^*(\tilde{\Delta})) \leq \tilde{\epsilon}$$

$\square$

*Proof.* (of Lemma 5) Assume $\tilde{E}$ happens. For each $j$, by triangle inequality, we have that $\rho_{S_j}(\hat{h}_j, h) \leq \operatorname{err}_{S_j}(\hat{h}_j) + \operatorname{err}_{S_j}(h)$. If $h \in V_j$, then, by defintion of $V_j$,

$$\operatorname{err}_{S_j}(h) - \operatorname{err}_{S_j}(\hat{h}_j) \leq \frac{\tilde{\epsilon}}{2} + \sigma(n_j, \tilde{\delta}_j) + \sqrt{\sigma(n_j, \tilde{\delta}_j)\operatorname{err}_{S_j}(\hat{h}_j)} + \sqrt{\sigma(n_j, \tilde{\delta}_j)\operatorname{err}_{S_j}(h)}$$

Using the fact that $A \leq B + C\sqrt{A} \Rightarrow A \leq 2B + C^2$,

$$\operatorname{err}_{S_j}(h) \leq \tilde{\epsilon} + 2\operatorname{err}_{S_j}(\hat{h}_j) + 2\sqrt{\sigma(n_j, \tilde{\delta}_j)\operatorname{err}_{S_j}(\hat{h}_j)} + 3\sigma(n_j, \tilde{\delta}_j) \leq 3\operatorname{err}_{S_j}(\hat{h}_j) + 4\sigma(n_j, \tilde{\delta}_j) + \tilde{\epsilon}$$

Since

$$\operatorname{err}_{S_j}(\hat{h}_j) \leq \operatorname{err}_{S_j}(h^*(\tilde{\Delta})) \leq \nu^*(\tilde{\Delta}) + \sqrt{\sigma(n_j, \tilde{\delta}_j)\nu^*(\tilde{\Delta})} + \sigma(n_j, \tilde{\delta}_j) \leq 2\nu^*(\tilde{\Delta}) + 2\sigma(n_j, \tilde{\delta}_j),$$

by the triangle inequality, we get that for all $h \in V_j$,

$$\rho_{S_j}(h, \hat{h}_j) \leq \operatorname{err}_{S_j}(h) + \operatorname{err}_{S_j}(\hat{h}_j) \leq 8\nu^*(\tilde{\Delta}) + 12\sigma(n_j, \tilde{\delta}_j) + \tilde{\epsilon} \tag{18}$$

Now observe that for any $j$,

$$\sup_{h \in V_j} \sqrt{\sigma(n_j, \tilde{\delta}_j)\rho_{S_j}(h, \hat{h}_j)} + \sigma(n_j, \tilde{\delta}_j)$$

$$\leq \sup_{h \in V_j} \max(2\sqrt{\sigma(n_j, \tilde{\delta}_j)\rho_{S_j}(h, \hat{h}_j)}, 2\sigma(n_j, \tilde{\delta}_j))$$

$$\leq \max(2\sqrt{(8\nu^*(\tilde{\Delta}) + 12\sigma(n_j, \tilde{\delta}_j) + \tilde{\epsilon})\sigma(n_j, \tilde{\delta}_j)}, 2\sigma(n_j, \tilde{\delta}_j))$$

$$\leq \max(12\sqrt{2\nu^*(\tilde{\Delta})\sigma(n_j, \tilde{\delta}_j)}, \tilde{\epsilon}/6, 216\sigma(n_j, \tilde{\delta}_j)),$$

Where the first inequality follows from $A + B \leq 2\max(A, B)$, the second inequality follows from Equation (18), the third inequality follows from $\sqrt{A+B} \leq \sqrt{A} + \sqrt{B}$, $A + B + C \leq 3\max(A, B, C)$ and $\sqrt{AB} \leq \max(A, B)$.

It can be easily seen that there exists some constant $c_1 > 0$, such that taking $j_1 = \lceil \log\left(\frac{c_1}{2}(d \ln \frac{1}{\tilde{\epsilon}} + \ln \frac{1}{\tilde{\delta}})(\frac{\nu^*(\tilde{\Delta})+\tilde{\epsilon}}{\tilde{\epsilon}^2})\right) \rceil$ ensures that $n_{j_1} \geq \frac{c_1}{2}(d \ln \frac{1}{\tilde{\epsilon}} + \ln \frac{1}{\tilde{\delta}})(\frac{\nu^*(\tilde{\Delta})+\tilde{\epsilon}}{\tilde{\epsilon}^2})$; this, in turn, suffices to make

$$\max(12\sqrt{2\nu^*(\tilde{\Delta})\sigma(n_j, \tilde{\delta}_j)}, 216\sigma(n_j, \tilde{\delta}_j)) \leq \tilde{\epsilon}/6$$

Hence the stopping criterion $\sup_{h \in V_j} \sqrt{\sigma(n_j, \tilde{\delta}_j)\rho_{S_j}(h, \hat{h}_j)} + \sigma(n_j, \tilde{\delta}_j) \leq \tilde{\epsilon}/6$ is satisfied in iteration $j_1$, and Algorithm 2 exits at iteration $j_0 \leq j_1$, which ensures that $n_{j_0} \leq n_{j_1} \leq c_1(d \ln \frac{1}{\tilde{\epsilon}} + \ln \frac{1}{\tilde{\delta}})(\frac{\nu^*(\tilde{\Delta})+\tilde{\epsilon}}{\tilde{\epsilon}^2})$. $\square$

The following lemma examines the behavior of Algorithm 2 under the Tsybakov Noise Condition and is crucial in the proof of Theorem 5. We observe that even if the $(C_0, \kappa)$-Tsybakov Noise Conditions hold with respect to $D$, they do not necessarily hold with respect to $\Gamma_k$. In particular, it is not necessarily true that:

$$\rho_{\tilde{\Gamma}_k}(h, h^*(D)) \leq C_0(\text{err}_{\tilde{\Gamma}_k}(h) - \text{err}_{\tilde{\Gamma}_k}(h^*(D)))^{\frac{1}{\kappa}}, \forall h \in V_k$$

However, we show that an "approximate" Tsybakov Noise Condition with a significantly larger "$C_0$", namely Condition (19) is met by $\tilde{\Gamma}_k$ and $V_k$, with $C = \max(8C_0, 4)\phi_k^{\frac{1}{\kappa}-1}$ and $\tilde{h} = h^*(D)$. In the Lemma below, we carefully track the dependence of the number of our label queries on $C$, since $C = \max(8C_0, 4)\phi_k^{\frac{1}{\kappa}-1}$ can be $\omega(1)$ in our particular application.

**Lemma 11.** *Suppose we run Algorithm 2 with inputs hypothesis set $V$, example distribution $\tilde{\Delta}$, labelling oracle $\mathcal{O}$, excess generalization error $\tilde{\epsilon}$ and confidence $\tilde{\delta}$. Then there exists some absolute constant $c_2 > 0$ (independent of $C$) such that the following holds. Suppose there exist $C > 0$ and a classifier $\tilde{h} \in V$, such that*

$$\forall h \in V, \rho_{\tilde{\Delta}}(h, \tilde{h}) \leq C \max(\tilde{\epsilon}, err_{\tilde{\Delta}}(h) - err_{\tilde{\Delta}}(\tilde{h}))^{\frac{1}{\kappa}}, \tag{19}$$

*where $\tilde{\epsilon}$ is the target exccess error parameter in Algorithm 2. Then, on the event that Algorithm 2 succeeds,*

$$n_{j_0} \leq c_2 \max((d\ln\frac{1}{\tilde{\epsilon}} + \ln\frac{1}{\tilde{\delta}})\tilde{\epsilon}^{-1}, (d\ln(C\tilde{\epsilon}^{\frac{1}{\kappa}-2}) + \ln\frac{1}{\tilde{\delta}})C\tilde{\epsilon}^{\frac{1}{\kappa}-2})$$

Observe that Condition (19), the approximate Tsybakov Noise Condition in the statement of Lemma 11, is with respect to $\tilde{h}$, which is not necessarily the true risk minimizer in $V$ with respect to $\tilde{\Delta}$. We therefore prove Lemma 11 in three steps; first, in Lemma 12, we analyze the difference $\text{err}_{\tilde{\Delta}}(\hat{h}) - \text{err}_{\tilde{\Delta}}(\tilde{h})$, where $\hat{h}$ is the empirical risk minimizer. Then, in Lemma 13, we bound the difference $\text{err}_{\tilde{\Delta}}(h) - \text{err}_{\tilde{\Delta}}(\tilde{h})$ for any $h \in V_j$ for some $j$. Finally, we combine these two lemmas to provide sample complexity bounds for the $V_{j_0}$ output by Algorithm 2.

*Proof.* (of Lemma 11) Assume the event $\tilde{E}$ happens. Then,

Consider iteration $j$, by Lemma 13, if $h \in V_j$, then

$$\rho_{\tilde{\Delta}}(h, \hat{h}_j) \leq \rho_{\tilde{\Delta}}(h, \tilde{h}) + \rho_{\tilde{\Delta}}(\hat{h}_j, \tilde{h}) \leq \max(2C(36\tilde{\epsilon})^{\frac{1}{\kappa}}, 2C(52\sigma(n_j, \tilde{\delta}_j))^{\frac{1}{\kappa}}, 2C(6400C\sigma(n_j, \tilde{\delta}_j))^{\frac{1}{2\kappa-1}}). \tag{20}$$

We can write:

$$\begin{aligned}
\sup_{h \in V_j} \sigma(n_j, \tilde{\delta}_j) + \sqrt{\sigma(n_j, \tilde{\delta}_j)\rho_{S_j}(h, \hat{h}_j)} &\leq \sup_{h \in V_j} 3\sigma(n_j, \tilde{\delta}_j) + \sqrt{2\sigma(n_j, \tilde{\delta}_j)\rho_{\tilde{\Delta}}(h, \hat{h}_j)} \\
&\leq \sup_{h \in V_j} \max(6\sigma(n_j, \tilde{\delta}_j), 2\sqrt{2\sigma(n_j, \tilde{\delta}_j)\rho_{\tilde{\Delta}}(h, \hat{h}_j)}),
\end{aligned}$$

where the first inequality follows from Equation (23) and the second inequality follows $A + B \leq 2\max(A, B)$. We can further use Equation (20) to show that this is at most:

$$\leq \max(6\sigma(n_j, \tilde{\delta}_j), (16C\sigma(n_j, \tilde{\delta}_j))^{\frac{1}{2}}(36\tilde{\epsilon})^{\frac{1}{2\kappa}}, (16C\sigma(n_j, \tilde{\delta}_j))^{\frac{1}{2}}(52\sigma(n_j, \tilde{\delta}_j))^{\frac{1}{2\kappa}}, (6400C\sigma(n_j, \tilde{\delta}_j))^{\frac{\kappa}{2\kappa-1}})$$

$$\leq \max(6\sigma(n_j, \tilde{\delta}_j), \tilde{\epsilon}/6, (6400C\sigma(n_j, \tilde{\delta}_j))^{\frac{\kappa}{2\kappa-1}})$$

Here the last inequality follows from the fact that $(16C\sigma(n_j, \tilde{\delta}_j))^{\frac{1}{2}}(36\tilde{\epsilon})^{\frac{1}{2\kappa}} \leq \max((3456C\sigma(n_j, \tilde{\delta}_j))^{\frac{\kappa}{2\kappa-1}}, \tilde{\epsilon}/6)$ and $(16C\sigma(n_j, \tilde{\delta}_j))^{\frac{1}{2}}(52\sigma(n_j, \tilde{\delta}_j))^{\frac{1}{2\kappa}} \leq \max((144C\sigma(n_j, \tilde{\delta}_j))^{\frac{\kappa}{2\kappa-1}}, 6\sigma(n_j, \tilde{\delta}_j))$, since $A^{\frac{2\kappa-1}{2\kappa}}B^{\frac{1}{2\kappa}} \leq \max(A, B)$.

It can be easily seen that there exists $c_2 > 0$, such that taking $j_1 = \lceil \log \frac{c_2}{2}(d\ln\frac{\max(C,1)}{\tilde{\epsilon}} + \ln\frac{1}{\tilde{\delta}})(C\tilde{\epsilon}^{\frac{1}{\kappa}-2} + \tilde{\epsilon}^{-1}) \rceil$, so that $n_j \geq \frac{c_2}{2}(d\ln\frac{\max(C,1)}{\tilde{\epsilon}} + \ln\frac{1}{\tilde{\delta}})(C\tilde{\epsilon}^{\frac{1}{\kappa}-2} + \tilde{\epsilon}^{-1})$ suffices to make

$$\max(6\sigma(n_j, \tilde{\delta}_j), (6400C\sigma(n_j, \tilde{\delta}_j))^{\frac{\kappa}{2\kappa-1}}) \leq \tilde{\epsilon}/6$$

Hence the stopping criterion $\sup_{h \in V_j} \sqrt{\sigma(n_j, \tilde{\delta}_j)\rho_{S_j}(h, \hat{h}_j)} + \sigma(n_j, \tilde{\delta}_j) \leq \tilde{\epsilon}/6$ is satisfied in iteration $j_1$. Thus the number of the exit iteration $j_0$ satisfies $j_0 \leq j_1$, and $n_{j_0} \leq n_{j_1} \leq c_2 \max((d \ln \frac{1}{\tilde{\epsilon}} + \ln \frac{1}{\delta})\tilde{\epsilon}^{-1}, (d \ln(C\tilde{\epsilon}^{\frac{1}{\kappa}-2}) + \ln \frac{1}{\delta})C\tilde{\epsilon}^{\frac{1}{\kappa}-2})$.

$\square$

**Lemma 12.** *Suppose there exist $C > 0$ and a classifier $\tilde{h} \in V$, such that Equation (19) holds. Suppose we draw a set $S$ of $n$ examples, denote the empirical risk minimizer over $S$ as $\hat{h}$, then with probability $1 - \delta$:*

$$err_{\tilde{\Delta}}(\hat{h}) - err_{\tilde{\Delta}}(\tilde{h}) \leq \max(2\sigma(n, \delta), (4C\sigma(n, \delta))^{\frac{\kappa}{2\kappa-1}}, 2\tilde{\epsilon})$$

$$\rho_{\tilde{\Delta}}(\hat{h}, \tilde{h}) \leq \max(C(2\sigma(n, \delta))^{\frac{1}{\kappa}}, C(4C\sigma(n, \delta))^{\frac{1}{2\kappa-1}}, C(2\tilde{\epsilon})^{\frac{1}{\kappa}})$$

*Proof.* By Lemma 7, with probability $1 - \delta$, Equation (6) holds. Assume this happens.

$$\begin{aligned}
&\text{err}_{\tilde{\Delta}}(\hat{h}) - \text{err}_{\tilde{\Delta}}(\tilde{h}) \\
\leq\quad & \sigma(n, \delta) + \sqrt{\sigma(n, \delta)\rho_{\tilde{\Delta}}(\hat{h}, \tilde{h})} \\
\leq\quad & 2 \max(\sigma(n, \delta), \sqrt{\sigma(n, \delta)C(\text{err}_{\tilde{\Delta}}(h) - \text{err}_{\tilde{\Delta}}(\tilde{h})^{\frac{1}{\kappa}})}, \sqrt{\sigma(n, \delta)C\tilde{\epsilon}^{\frac{1}{\kappa}}}) \\
\leq\quad & \max(2\sigma(n, \delta), (4C\sigma(n, \delta))^{\frac{\kappa}{2\kappa-1}}, 2\tilde{\epsilon})
\end{aligned}$$

Where the first inequality is by Equation (6) of Lemma 7; the second inequality follow from Equation (19) and $A + B \leq 2\max(A, B)$. The third inequality follows from $2\sqrt{\sigma(n, \delta)C\tilde{\epsilon}^{\frac{1}{\kappa}}} \leq \max(2(C\sigma(n, \delta))^{\frac{2\kappa-1}{2\kappa}}, 2\tilde{\epsilon})$, since $A^{\frac{2\kappa-1}{2\kappa}}B^{\frac{1}{2\kappa}} \leq \max(A, B)$. As a consequence, by Equation (19),

$$\rho_{\tilde{\Delta}}(\hat{h}, \tilde{h}) \leq \max(C(2\sigma(n, \delta))^{\frac{1}{\kappa}}, C(4C\sigma(n, \delta))^{\frac{1}{2\kappa-1}}, C(2\tilde{\epsilon})^{\frac{1}{\kappa}})$$

$\square$

**Lemma 13.** *Suppose there exist a $C > 0$ and a classifier $\tilde{h} \in V$ such that Equation (19) holds. Suppose we draw a set $S$ of $n$ iid examples, and let $\hat{h}$ denote the empirical risk minimizer over $S$. Moreover, we define:*

$$\tilde{V} = \left\{ h \in V : err_S(h) \leq err_S(\hat{h}) + \frac{\tilde{\epsilon}}{2} + \sigma(n, \delta) + \sqrt{\sigma(n, \delta)\rho_S(h, \hat{h})} \right\}$$

*then with probability $1 - \delta$, for all $h \in \tilde{V}$,*

$$err_{\tilde{\Delta}}(h) - err_{\tilde{\Delta}}(\tilde{h}) \leq \max(52\sigma(n, \delta), 36\tilde{\epsilon}, (6400C\sigma(n, \delta))^{\frac{\kappa}{2\kappa-1}})$$

$$\rho_{\tilde{\Delta}}(h, \tilde{h}) \leq \max(C(36\tilde{\epsilon})^{\frac{1}{\kappa}}, C(52\sigma(n, \delta))^{\frac{1}{\kappa}}, C(6400C\sigma(n, \delta))^{\frac{1}{2\kappa-1}})$$

*Proof.* First, by Lemma 12,

$$\text{err}_{\tilde{\Delta}}(\hat{h}) - \text{err}_{\tilde{\Delta}}(\tilde{h}) \leq \max(2\sigma(n, \delta), (4C\sigma(n, \delta))^{\frac{\kappa}{2\kappa-1}}, 2\tilde{\epsilon}) \qquad (21)$$

$$\rho_{\tilde{\Delta}}(\hat{h}, \tilde{h}) \leq \max(C(2\sigma(n, \delta))^{\frac{1}{\kappa}}, C(4C\sigma(n, \delta))^{\frac{1}{2\kappa-1}}, C(2\tilde{\epsilon})^{\frac{1}{\kappa}}) \qquad (22)$$

Next, if $h \in \tilde{V}$, then

$$\text{err}_S(h) - \text{err}_S(\hat{h}) \leq \sigma(n, \delta) + \sqrt{\sigma(n, \delta)\rho_S(h, \hat{h})} + \frac{\tilde{\epsilon}}{2}$$

Combining it with Equation (6) of Lemma 7: $\text{err}_{\tilde{\Delta}}(h) - \text{err}_{\tilde{\Delta}}(\hat{h}) \leq \text{err}_S(h) - \text{err}_S(\hat{h}) + \sqrt{\sigma(n, \delta)\rho_S(h, \hat{h})} + \sigma(n, \delta)$, we get

$$\text{err}_{\tilde{\Delta}}(h) - \text{err}_{\tilde{\Delta}}(\hat{h}) \leq 2\sigma(n, \delta) + 2\sqrt{\sigma(n, \delta)\rho_S(h, \hat{h})} + \frac{\tilde{\epsilon}}{2}$$

By Equation (7) of Lemma 7,

$$\rho_S(h,\hat{h}) \le \rho_{\tilde{\Delta}}(h,\hat{h}) + \sqrt{\sigma(n,\delta)\rho_{\tilde{\Delta}}(h,\hat{h})} + \sigma(n,\delta) \le 2\rho_{\tilde{\Delta}}(h,\hat{h}) + 2\sigma(n,\delta) \qquad (23)$$

Therefore,

$$\mathrm{err}_{\tilde{\Delta}}(h) - \mathrm{err}_{\tilde{\Delta}}(\hat{h}) \le 5\sigma(n,\delta) + 3\sqrt{\sigma(n,\delta)\rho_{\tilde{\Delta}}(h,\hat{h})} + \frac{\tilde{\epsilon}}{2} \qquad (24)$$

Hence

$$\begin{aligned}
&\mathrm{err}_{\tilde{\Delta}}(h) - \mathrm{err}_{\tilde{\Delta}}(\tilde{h}) \\
={}& (\mathrm{err}_{\tilde{\Delta}}(h) - \mathrm{err}_{\tilde{\Delta}}(\hat{h})) + (\mathrm{err}_{\tilde{\Delta}}(\hat{h}) - \mathrm{err}_{\tilde{\Delta}}(\tilde{h})) \\
\le{}& (4C\sigma(n,\delta))^{\frac{\kappa}{2\kappa-1}} + 7\sigma(n,\delta) + 3\tilde{\epsilon} + 3\sqrt{\sigma(n,\delta)\rho_{\tilde{\Delta}}(h,\hat{h})} \\
\le{}& (4C\sigma(n,\delta))^{\frac{\kappa}{2\kappa-1}} + 7\sigma(n,\delta) + 3\tilde{\epsilon} + 3\sqrt{\sigma(n,\delta)\rho_{\tilde{\Delta}}(h,\tilde{h})} + 3\sqrt{\sigma(n,\delta)\rho_{\tilde{\Delta}}(\tilde{h},\hat{h})}
\end{aligned}$$

Here the first inequality follows from Equations (21) and (24) and $\max(A,B,C) \le A + B + C$, and the second inequality follows from triangle inequality and $\sqrt{A+B} \le \sqrt{A} + \sqrt{B}$.

From Equation (22), $\sigma(n,\delta)\rho_{\tilde{\Delta}}(\hat{h},\tilde{h})$ is at most:

$$\begin{aligned}
\le{}& C\sigma(n,\delta) \cdot ((2\tilde{\epsilon})^{1/\kappa} + (2\sigma(n,\delta))^{1/\kappa} + (4C\sigma(n,\delta))^{1/(2\kappa-1)}) \\
\le{}& (4C\sigma(n,\delta))^{2\kappa/(2\kappa-1)} + C\sigma(n,\delta)((2\tilde{\epsilon})^{1/\kappa} + (2\sigma(n,\delta))^{1/\kappa}) \\
\le{}& (4C\sigma(n,\delta))^{2\kappa/(2\kappa-1)} + \max(4\tilde{\epsilon}^2, (C\sigma(n,\delta))^{2\kappa/(2\kappa-1)}) + \max(4\sigma(n,\delta)^2, (C\sigma(n,\delta))^{2\kappa/(2\kappa-1)}),
\end{aligned}$$

where the first step follows from Equation (22), the second step from algebra, and the third step from using the fact that $A^{\frac{2\kappa-1}{\kappa}} B^{\frac{1}{\kappa}} \le \max(A^2, B^2)$. Plugging this in to the previous equation, and using $\max(A,B) \le A + B$ and $\sqrt{A+B} \le \sqrt{A} + \sqrt{B}$, we get that:

$$\mathrm{err}_{\tilde{\Delta}}(h) - \mathrm{err}_{\tilde{\Delta}}(\tilde{h}) \le 10(4C\sigma(n,\delta))^{\kappa/(2\kappa-1)} + 9\tilde{\epsilon} + 13\sigma(n,\delta) + 3\sqrt{\sigma(n,\delta)\rho_{\tilde{\Delta}}(h,\tilde{h})}$$

Combining this with the fact that $A + B + C + D \le 4\max(A,B,C,D)$, we get that this is at most:

$$\le \max(40(4C\sigma(n,\delta))^{\kappa/(2\kappa-1)}, 36\tilde{\epsilon}, 52\sigma(n,\delta), 12\sqrt{\sigma(n,\delta)\rho_{\tilde{\Delta}}(h,\tilde{h})})$$

Combining this with Condition (19), we get that this is at most:

$$\max(40(4C\sigma(n,\delta))^{\kappa/(2\kappa-1)}, 36\tilde{\epsilon}, 52\sigma(n,\delta), 12\sqrt{C\sigma(n,\delta)\tilde{\epsilon}^{1/\kappa}}, 12\sqrt{C\sigma(n,\delta)(\mathrm{err}_{\tilde{\Delta}}(h) - \mathrm{err}_{\tilde{\Delta}}(\tilde{h}))^{1/\kappa}})$$

Using $A^{(2\kappa-1)/2\kappa} B^{1/2\kappa} \le \max(A,B)$, we get that $\sqrt{C\sigma(n,\delta)\tilde{\epsilon}^{1/\kappa}} \le \max(\tilde{\epsilon}, (C\sigma(n,\delta))^{\kappa/(2\kappa-1)})$. Also note $\mathrm{err}_{\tilde{\Delta}}(h) - \mathrm{err}_{\tilde{\Delta}}(\tilde{h}) \le 12\sqrt{C\sigma(n,\delta)(\mathrm{err}_{\tilde{\Delta}}(h) - \mathrm{err}_{\tilde{\Delta}}(\tilde{h}))^{1/\kappa}}$ implies $\mathrm{err}_{\tilde{\Delta}}(h) - \mathrm{err}_{\tilde{\Delta}}(\tilde{h}) \le (144C\sigma(n,\delta))^{\kappa/(2\kappa-1)}$. Thus we have

$$\mathrm{err}_{\tilde{\Delta}}(h) - \mathrm{err}_{\tilde{\Delta}}(\tilde{h}) \le \max(36\tilde{\epsilon}, 52\sigma(n,\delta), (6400C\sigma(n,\delta))^{\frac{\kappa}{2\kappa-1}})$$

Invoking (19) again, we have that:

$$\rho_{\tilde{\Delta}}(h,\tilde{h}) \le \max(C(36\tilde{\epsilon})^{\frac{1}{\kappa}}, C(52\sigma(n,\delta))^{\frac{1}{\kappa}}, C(6400C\sigma(n,\delta))^{\frac{1}{2\kappa-1}})$$

$\square$

# D    Remaining Proofs from Section 2

*Proof.* (Of Lemma 1) Assuming $E_r$ happens, we prove the lemma by induction.
**Base Case:** For $k = 1$, clearly $h^*(D) \in V_1 = \mathcal{H}$.
**Inductive Case:** Assume $h^*(D) \in V_k$. As we are in the realizable case, $h^*(D)$ is consistent with the examples $S_k$ drawn in Step 8 of Algorithm 1; thus $h^*(D) \in V_{k+1}$. The lemma follows.    $\square$

*Proof.* (Of Lemma 2) We use $\tilde{h}_k = \operatorname{argmin}_{h \in V_k} \operatorname{err}_{\tilde{\Gamma}_k}(h)$ to denote the optimal classifier in $V_k$ with respect to the distribution $\tilde{\Gamma}_k$. Assuming $E_a$ happens, we prove the lemma by induction.

**Base Case:** For $k = 1$, clearly $h^*(D) \in V_1 = \mathcal{H}$.

**Inductive Case:** Assume $h^* \in V_k$. In order to show the inductive case, our goal is to show that:

$$\mathbb{P}_{\tilde{\Gamma}_k}(h^*(D)(x) \neq y) - \mathbb{P}_{\tilde{\Gamma}_k}(\tilde{h}_k(x) \neq y) \leq \frac{\epsilon_k}{16\phi_k} \tag{25}$$

If (25) holds, then, by (2.1) of Lemma 4, we know that if Algorithm 2 succeeds when called in iteration $k$ of Algorithm 1, then, it is guaranteed that $h^* \in V_{k+1}$.

We therefore focus on showing (25). First, from Equation (12) of Lemma 9, we have:

$$(\operatorname{err}_{\tilde{U}_k}(h^*(D)) - \operatorname{err}_{\tilde{U}_k}(\tilde{h}_k)) - (\operatorname{err}_D(h^*(D)) - \operatorname{err}_D(\tilde{h}_k)) \leq \frac{\epsilon_k}{32}$$

As $\operatorname{err}_D(h^*(D)) \leq \operatorname{err}_D(\tilde{h}_k)$, we get:

$$\operatorname{err}_{\tilde{U}_k}(h^*(D)) \leq \operatorname{err}_{\tilde{U}_k}(\tilde{h}_k) + \frac{\epsilon_k}{32} \tag{26}$$

On the other hand, by Equation (14) of Lemma 10 and triangle inequality,

$$\mathbb{E}_{\tilde{U}_k}[I(\tilde{h}_k(x) \neq y)(1 - \gamma_k(x))] - \mathbb{E}_{\tilde{U}_k}[I(h^*(D)(x) \neq y)(1 - \gamma_k(x))] \tag{27}$$

$$\leq \quad \mathbb{E}_{\tilde{U}_k}[I(h^*(D)(x) \neq \tilde{h}_k(x))(1 - \gamma_k(x))] \leq \frac{\epsilon_k}{32} \tag{28}$$

Combining Equations (26) and (27), we get:

$$
\begin{aligned}
\mathbb{E}_{\tilde{U}_k}[I(h^*(D)(x) \neq y)\gamma_k(x)] &= \operatorname{err}_{\tilde{U}_k}(h^*(D)(x)) - \mathbb{E}_{\tilde{U}_k}[I(h^*(D)(x) \neq y)(1 - \gamma_k(x))] \\
&\leq \operatorname{err}_{\tilde{U}_k}(\tilde{h}_k(x)) + \epsilon_k/32 - \mathbb{E}_{\tilde{U}_k}[I(h^*(D)(x) \neq y)(1 - \gamma_k(x))] \\
&\leq \mathbb{E}_{\tilde{U}_k}[I(\tilde{h}_k(x) \neq y)\gamma_k(x)] + \mathbb{E}_{\tilde{U}_k}[I(\tilde{h}(x) \neq y)(1 - \gamma_k(x))] + \epsilon_k/32 \\
&\quad - \mathbb{E}_{\tilde{U}_k}[I(h^*(D)(x) \neq y)(1 - \gamma_k(x))] \\
&\leq \mathbb{E}_{\tilde{U}_k}[I(\tilde{h}_k(x) \neq y)\gamma_k(x)] + \epsilon_k/16
\end{aligned}
$$

Dividing both sides by $\phi_k$, we get:

$$\mathbb{P}_{\tilde{\Gamma}_k}(h^*(D)(x) \neq y) - \mathbb{P}_{\tilde{\Gamma}_k}(\tilde{h}_k(x) \neq y) \leq \frac{\epsilon_k}{16\phi_k},$$

from which the lemma follows. $\qquad \square$

*Proof.* (of Lemma 3) Assuming $E_r$ happens, we prove the lemma by induction.

**Base Case:** For $k = 1$, clearly $\operatorname{err}_D(h) \leq 1 \leq \epsilon_1 = \epsilon 2^{k_0}, \forall h \in V_1 = \mathcal{H}$.

**Inductive Case:** Note that $\forall h, h' \in V_{k+1} \subseteq V_k$, by Equation (14) of Lemma 10, we have:

$$\mathbb{E}_{\tilde{U}_k}[I(h(x) \neq h'(x))(1 - \gamma_k(x))] \leq \frac{\epsilon_k}{8}$$

By the proof of Lemma 1, $h^*(D) \in V_{k+1}$ on event $E_r$, thus $\forall h \in V_{k+1}$,

$$\mathbb{E}_{\tilde{U}_k}[I(h(x) \neq h^*(D)(x))(1 - \gamma_k(x))] \leq \frac{\epsilon_k}{8} \tag{29}$$

Since any $h \in V_{k+1}$, $h$ is consistent with $S_k$ of size $m_k = \frac{1536\phi_k}{\epsilon_k}(d \ln \frac{1536\phi_k}{\epsilon_k} + \ln \frac{48}{\delta_k})$, we have that for all $h \in V_{k+1}$,

$$\mathbb{P}_{\tilde{\Gamma}_k}(h(x) \neq h^*(D)(x)) \leq \frac{\epsilon_k}{8\phi_k}$$

That is,

$$\mathbb{E}_{\tilde{U}_k}[I(h(x) \neq h^*(D)(x))\gamma_k(x)] \leq \frac{\epsilon_k}{8}$$

Combining this with Equation (29) above,

$$\mathbb{P}_{\tilde{U}_k}(h(x) \neq h^*(D)(x)) \leq \frac{\epsilon_k}{4}$$

By Equation (11) of Lemma 9,

$$\mathbb{P}_D(h(x) \neq h^*(D)(x)) \leq \frac{\epsilon_k}{2} = \epsilon_{k+1}$$

The lemma follows. $\qquad \square$

*Proof.* (of Lemma 6) Assuming $E_a$ happens, we prove the lemma by induction.

**Base Case:** For $k = 1$, clearly $\text{err}_D(h) - \text{err}_D(h^*(D)) \leq 1 \leq \epsilon_1 = \epsilon 2^{k_0}, \forall h \in V_1 = \mathcal{H}$.

**Inductive Case:** Note that $\forall h, h' \in V_{k+1} \subseteq V_k$, by Equation (14) of Lemma 10,

$$\mathbb{E}_{\tilde{U}_k}[I(h(x) \neq y)(1-\gamma_k(x))] - \mathbb{E}_{\tilde{U}_k}[I(h'(D)(x) \neq y)(1-\gamma_k(x))] \leq \mathbb{E}_{\tilde{U}_k}[I(h(x) \neq h'(D)(x))(1-\gamma_k(x))] \leq \frac{\epsilon_k}{8}$$

From Lemma 2, $h^*(D) \in V_k$ whenever the event $E_a$ happens. Thus $\forall h \in V_{k+1}$,

$$\mathbb{E}_{\tilde{U}_k} I(h(x) \neq y)(1 - \gamma_k(x)) - \mathbb{E}_{\tilde{U}_k} I(h^*(D)(x) \neq y)(1 - \gamma_k(x)) \leq \frac{\epsilon_k}{8} \qquad (30)$$

On the other hand, if Algorithm 2 succeeds with target excess error $\frac{\epsilon_k}{8\phi_k}$, by item(2.2) of Lemma 4, for any $h \in V_{k+1}$,

$$\mathbb{P}_{\tilde{\Gamma}_k}(h(x) \neq y) - \min_{h \in V_k} \mathbb{P}_{\tilde{\Gamma}_k}(h(x) \neq y) \leq \frac{\epsilon_k}{8\phi_k}$$

Moreover, as $h^*(D) \in V_k$ from Lemma 2,

$$\mathbb{P}_{\tilde{\Gamma}_k}(h(x) \neq y) - \mathbb{P}_{\tilde{\Gamma}_k}(h^*(D)(x) \neq y) \leq \frac{\epsilon_k}{8\phi_k}$$

In other words,

$$\mathbb{E}_{\tilde{U}_k}[I(h(x) \neq y)\gamma_k(x)] - \mathbb{E}_{\tilde{U}_k}[I(h^*(D)(x) \neq y)\gamma_k(x)] \leq \frac{\epsilon_k}{8}$$

Combining this with Equation (30), we get that for all $h \in V_{k+1}$,

$$\mathbb{P}_{\tilde{U}_k}(h(x) \neq y) - \mathbb{P}_{\tilde{U}_k}(h^*(D)(x) \neq y) \leq \frac{\epsilon_k}{4}$$

Finally, combining this with Equation (12) of Lemma 9, we have that:

$$\mathbb{P}_D(h(x) \neq y) - \mathbb{P}_D(h^*(D)(x) \neq y) \leq \frac{\epsilon_k}{2} = \epsilon_{k+1}$$

The lemma follows. $\qquad \square$

*Proof.* (of Theorem 1) In the realizable case, We observe that for example $z_i$, $\zeta_i = \mathbb{P}(P(z_i) = -1)$, $\xi_i = \mathbb{P}(P(z_i) = 1)$, and $\gamma_i = \mathbb{P}(P(z_i) = 0)$. Suppose $h^* \in \mathcal{H}$ is the true hypothesis which has 0 error with respect to the data distribution. By the realizability assumption, $h^* \in V$. Moreover, $\mathbb{P}_U(P(x) \neq h^*(x), P(x) \neq 0) = \frac{1}{m}(\sum_{i:h^*(z_i)=+1} \zeta_i + \sum_{i:h^*(z_i)=-1} \xi_i) \leq \eta$ by Algorithm 3.

In the non-realizable case, we still have $\mathbb{P}_{x \sim U}(h^*(x) \neq P(x), P(x) \neq 0) \leq \eta$, hence by triangle inequality, $\mathbb{P}_{x \sim U}(P(x) \neq x, P(x) \neq 0) - \mathbb{P}_{x \sim U}(h^*(x) \neq y, P(x) \neq 0) \leq \eta$. Thus

$$\mathbb{P}_{x \sim U}(P(x) \neq y, P(x) \neq 0) \leq \mathbb{P}_{x \sim U}(h^*(x) \neq y) + \eta$$

$\qquad \square$

*Proof.* (of Theorem 2) Suppose $P'$ assigns probabilities $\{[\xi_i', \zeta_i', \gamma_i'], i = 1, \ldots, m\}$ to the unlabelled examples $z_i$, and suppose for the sake of contradiction that $\sum_{i=1}^m \xi_i' + \zeta_i' > \sum_{i=1}^m \xi_i + \zeta_i$. Then, $\{\xi_i', \zeta_i', \gamma_i'\}$'s cannot satisfy the LP in Algorithm 3, and thus there exists some $h' \in V$ for which constraint (2) is violated. The true hypothesis that generates the data could be any $h \in V$; if this true hypothesis is $h'$, then $\mathbb{P}_{x \sim U}(P'(x) \neq h'(x), P'(x) \neq 0) > \delta$. $\qquad \square$

# E  Proofs from Section 3

*Proof.* (of Theorem 4)

(1) In the realizable case, suppose that event $E_r$ happens. Then from Equation (15) of Lemma 10, while running Algorithm 3, we have that:

$$\phi_k \leq \mathbf{\Phi}_D(V_k, \frac{\epsilon_k}{128}) + \frac{\epsilon_k}{256} \leq \mathbf{\Phi}_D(B_D(h^*, \epsilon_k), \frac{\epsilon_k}{128}) + \frac{\epsilon_k}{256} \leq \mathbf{\Phi}_D(B_D(h^*, \epsilon_k), \frac{\epsilon_k}{256}) = \phi(\epsilon_k, \frac{\epsilon_k}{256})$$

where the second inequality follows from the fact that $V_k \subseteq B_D(h^*(D), \epsilon_k)$, and third inequality follows from Lemma 18 and denseness assuption.

Thus, there exists $c_3 > 0$ such that, in round $k$,

$$m_k = (d \ln \frac{1536\phi_k}{\epsilon_k} + \ln \frac{48}{\delta_k}) \frac{1536\phi_k}{\epsilon_k} \leq c_3 (d \ln \frac{\phi(\epsilon_k, \epsilon_k/256)}{\epsilon_k} + \ln(\frac{k_0 - k + 1}{\delta})) \frac{\phi(\epsilon_k, \epsilon_k/256)}{\epsilon_k}$$

Hence the total number of labels queried by Algorithm 1 is at most

$$\sum_{k=1}^{\lceil \log \frac{1}{\epsilon} \rceil} m_k \leq c_3 \sum_{k=1}^{\lceil \log \frac{1}{\epsilon} \rceil} (d \ln \frac{\phi(\epsilon_k, \epsilon_k/256)}{\epsilon_k} + \ln(\frac{k_0 - k + 1}{\delta})) \frac{\phi(\epsilon_k, \epsilon_k/256)}{\epsilon_k}$$

(2) In the agnostic case, suppose the event $E_a$ happens.
First, given $E_a$, from Equation (15) of Lemma 10 when running Algorithm 3,

$$\phi_k \leq \mathbf{\Phi}_D(V_k, \frac{\epsilon_k}{128}) + \frac{\epsilon_k}{256} \leq \mathbf{\Phi}_D(B_D(h^*, 2\nu^*(D) + \epsilon_k), \frac{\epsilon_k}{256}) = \phi(2\nu^*(D) + \epsilon_k, \frac{\epsilon_k}{256}) \quad (31)$$

where the second inequality follows from the fact that $V_k \subseteq B_D(h^*(D), 2\nu^*(D) + \epsilon_k)$ and the third inequality follows from Lemma 18 and denseness assumption.
Second, recall that $\tilde{h}_k = \operatorname{argmin}_{h \in V_k} \operatorname{err}_{\tilde{\Gamma}_k}(h)$,

$$
\begin{aligned}
\operatorname{err}_{\tilde{\Gamma}_k}(\tilde{h}_k) &= \min_{h \in V_k} \operatorname{err}_{\tilde{\Gamma}_k}(h) \\
&\leq \operatorname{err}_{\tilde{\Gamma}_k}(h^*(D)) \\
&= \frac{\mathbb{E}_{\tilde{U}_k}[I(h^*(D)(x) \neq y)\gamma_k(x)]}{\phi_k} \\
&\leq \frac{\mathbb{P}_{\tilde{U}_k}(h^*(D)(x) \neq y)}{\phi_k} \\
&\leq \frac{\nu^*(D) + \epsilon_k/64}{\phi_k}
\end{aligned}
$$

Here the first inequality follows from the suboptimality of $h^*(D)$ under distribution $\tilde{\Gamma}_k$, the second inequality follows from $\gamma_k(x) \leq 1$, and the third inequality follows from Equation (11).
Thus, conditioned on $E_a$, in iteration $k$, Algorithm 2 succeeds by Lemma 5, and there exists a constant $c_4 > 0$ such that the number of labels queried is

$$
\begin{aligned}
m_k &\leq c_1 \frac{\frac{\epsilon_k}{8\phi_k} + \operatorname{err}_{\tilde{\Gamma}_k}(\tilde{h}_k)}{(\frac{\epsilon_k}{8\phi_k})^2} (d \ln \frac{1}{\frac{\epsilon_k}{8\phi_k}} + \ln \frac{2}{\delta_k}) \\
&\leq c_4 (d \ln \frac{\phi(2\nu^*(D) + \epsilon_k, \epsilon_k/256)}{\epsilon_k} + \ln(\frac{k_0 - k + 1}{\delta})) \frac{\phi(2\nu^*(D) + \epsilon_k, \epsilon_k/256)}{\epsilon_k} (1 + \frac{\nu^*(D)}{\epsilon_k})
\end{aligned}
$$

Here the last line follows from Equation (31). Hence the total number of examples queried is at most:

$$\sum_{k=1}^{\lceil \log \frac{1}{\epsilon} \rceil} m_k \leq c_4 \sum_{k=1}^{\lceil \log \frac{1}{\epsilon} \rceil} (d \ln \frac{\phi(2\nu^*(D) + \epsilon_k, \epsilon_k/256)}{\epsilon_k} + \ln(\frac{k_0 - k + 1}{\delta})) \frac{\phi(2\nu^*(D) + \epsilon_k, \epsilon_k/256)}{\epsilon_k} (1 + \frac{\nu^*(D)}{\epsilon_k})$$

$\square$

*Proof.* (of Theorem 5) Assume $E_a$ happens.
First, from Equation (15) of Lemma 10 when running Algorithm 3,

$$\phi_k \leq \mathbf{\Phi}_D(V_k, \frac{\epsilon_k}{128}) + \frac{\epsilon_k}{256} \leq \mathbf{\Phi}_D(B_D(h^*, C_0 \epsilon_k^{\frac{1}{\kappa}}), \frac{\epsilon_k}{128}) + \frac{\epsilon_k}{256} \leq \mathbf{\Phi}_D(B_D(h^*, C_0 \epsilon_k^{\frac{1}{\kappa}}), \frac{\epsilon_k}{256}) = \phi(C_0 \epsilon_k^{\frac{1}{\kappa}}, \frac{\epsilon_k}{256})$$
$$(32)$$

where the second inequality follows from the fact that $V_k \subseteq B_D(h^*(D), C_0 \epsilon_k^{\frac{1}{\kappa}})$, and the third inequality follows from Lemma 18 and denseness assumption.

Second, for all $h \in V_k$,

$$\phi_k \rho_{\tilde{\Gamma}_k}(h, h^*(D))$$
$$= \mathbb{E}_{\tilde{U}_k} I(h(x) \neq h^*(D)(x)) \gamma_k(x)$$
$$\leq \rho_{\tilde{U}_k}(h, h^*(D))$$
$$\leq \rho_D(h, h^*(D)) + \epsilon_k/32$$
$$\leq C_0(\text{err}_D(h) - \text{err}_D(h^*(D)))^{\frac{1}{\kappa}} + \epsilon_k/32$$
$$\leq C_0(\text{err}_{\tilde{U}_k}(h) - \text{err}_{\tilde{U}_k}(h^*(D)) + \epsilon_k/64)^{\frac{1}{\kappa}} + \epsilon_k/32$$
$$= C_0(\mathbb{E}_{\tilde{U}_k}[I(h(x) \neq y)\gamma_k(x)] - \mathbb{E}_{\tilde{U}_k}[I(h^*(D)(x) \neq y)\gamma_k(x)]$$
$$+ \mathbb{E}_{\tilde{U}_k}[I(h(x) \neq y)(1 - \gamma_k(x))] - \mathbb{E}_{\tilde{U}_k}[I(h^*(D)(x) \neq y)(1 - \gamma_k(x))] + \epsilon_k/16)^{\frac{1}{\kappa}} + \epsilon_k/32$$

Here the first inequality follows from $\gamma_k(x) \leq 1$, the second inequality follows from Equation (13) of Lemma 9, the third inequality follows from Definition 1 and the fourth inequality follows from Equation (12) of Lemma 9. The above can be upper bounded by:

$$\leq C_0(\mathbb{E}_{\tilde{U}_k}[I(h(x) \neq y)\gamma_k(x)] - \mathbb{E}_{\tilde{U}_k}[I(h^*(D)(x) \neq y)\gamma_k(x)] + \epsilon_k/16)^{\frac{1}{\kappa}} + \epsilon_k/32$$
$$\leq 2C_0(\mathbb{E}_{\tilde{U}_k}[I(h(x) \neq y)\gamma_k(x)] - \mathbb{E}_{\tilde{U}_k}[I(h^*(D)(x) \neq y)\gamma_k(x)])^{\frac{1}{\kappa}} + 2C_0(\epsilon_k/16)^{\frac{1}{\kappa}} + \epsilon_k/32$$
$$\leq \max(8C_0, 4) \max((\mathbb{E}_{\tilde{U}_k}[I(h(x) \neq y)\gamma_k(x)] - \mathbb{E}_{\tilde{U}_k}[I(h^*(D)(x) \neq y)\gamma_k(x)]), \frac{\epsilon_k}{16})^{\frac{1}{\kappa}}$$
$$= \max(8C_0, 4)(\phi_k)^{\frac{1}{\kappa}} \max(\mathbb{P}_{\tilde{\Gamma}_k}(h(x) \neq y) - \mathbb{P}_{\tilde{\Gamma}_k}(h^*(D)(x) \neq y), \frac{\epsilon_k}{8\phi_k})^{\frac{1}{\kappa}}$$

Here the first inequality follows from Equation (14) of Lemma 10 and triangle inequality $\mathbb{E}_{\tilde{U}_k}[I(h(x) \neq y)\gamma_k(x)] - \mathbb{E}_{\tilde{U}_k}[I(h^*(D)(x) \neq y)\gamma_k(x)] \leq \mathbb{E}_{\tilde{U}_k}[I(h(x) \neq h^*(D)(x))\gamma_k(x)] \leq \epsilon_k/32$, and the last two inequalities follow from simple algebra.

Dividing both sides by $\phi_k$, we get:

$$\rho_{\tilde{\Gamma}_k}(h, h^*(D)) \leq C_1(\phi_k)^{\frac{1}{\kappa}-1} \max(\text{err}_{\tilde{\Gamma}_k}(h) - \text{err}_{\tilde{\Gamma}_k}(h^*(D)), \frac{\epsilon_k}{8\phi_k})^{\frac{1}{\kappa}}$$

where $C_1 = \max(8C_0, 4)$. Thus in iteration $k$, Condition (19) in Lemma 11 holds with $C := C_1(\phi_k)^{\frac{1}{\kappa}-1}$ and $\tilde{h} := h^*(D)$. Thus, from Lemma 11, Algorithm 2 succeeds, and there exists a constant $c_5 > 0$, such that the number of labels queried is

$$m_k \leq c_2 \max((d \ln(C_1(\phi_k)^{\frac{1}{\kappa}-1}(\frac{\epsilon_k}{8\phi_k})^{\frac{1}{\kappa}-2}) + \ln\frac{2}{\delta_k})(C_1(\phi_k)^{\frac{1}{\kappa}-1}(\frac{\epsilon_k}{8\phi_k})^{\frac{1}{\kappa}-2}),$$
$$(d\ln(\frac{\epsilon_k}{8\phi_k})^{-1} + \ln\frac{2}{\delta_k})(\frac{\epsilon_k}{8\phi_k})^{-1})$$
$$\leq c_5(d\ln(\phi_k \epsilon_k^{\frac{1}{\kappa}-2}) + \ln(\frac{k_0-k+1}{\delta}))\phi_k \epsilon_k^{\frac{1}{\kappa}-2}$$
$$\leq c_5(d\ln(\phi(C_0\epsilon_k^{\frac{1}{\kappa}}, \frac{\epsilon_k}{256})\epsilon_k^{\frac{1}{\kappa}-2}) + \ln(\frac{k_0-k+1}{\delta}))\phi(C_0\epsilon_k^{\frac{1}{\kappa}}, \frac{\epsilon_k}{256})\epsilon_k^{\frac{1}{\kappa}-2}$$

Where the last line follows from Equation (31). Hence the total number of examples queried is at most

$$\sum_{k=1}^{\lceil \log \frac{1}{\epsilon} \rceil} m_k \leq c_5 \sum_{k=1}^{\lceil \log \frac{1}{\epsilon} \rceil} (d\ln(\phi(C_0\epsilon_k^{\frac{1}{\kappa}}, \frac{\epsilon_k}{256})\epsilon_k^{\frac{1}{\kappa}-2}) + \ln(\frac{k_0-k+1}{\delta}))\phi(C_0\epsilon_k^{\frac{1}{\kappa}}, \frac{\epsilon_k}{256})\epsilon_k^{\frac{1}{\kappa}-2}$$

$\square$

The following lemma is an immediate corollary of Theorem 21, item (a) of Lemma 2 and Lemma 3 of [4]:

**Lemma 14.** *Suppose $D$ is isotropic and log-concave on $R^d$, and $\mathcal{H}$ is the set of homogeneous linear classifiers on $R^d$, then there exist absolute constants $c_6, c_7 > 0$ such that $\phi(r, \eta) \leq c_6 r \ln \frac{c_7 r}{\eta}$.*

*Proof.* (of Lemma 14) Denote $w_h$ as the unit vector $w$ such that $h(x) = \text{sign}(w \cdot x)$, and $\theta(w, w')$ to be the angle between vectors $w$ and $w'$. If $h \in B_D(h^*, r)$, then by Lemma 3 of [4], there exists some constant $c_{11} > 0$ such that $\theta(w_h, w_{h^*}) \leq \frac{r}{c_{11}}$. Also, by Lemma 21 of [4], there exists some constants $c_{12}, c_{13} > 0$, such that, if $\theta(w, w') = \alpha$ then

$$\mathbb{P}_D(\text{sign}(w \cdot x) \neq \text{sign}(w' \cdot x), |w \cdot x| \geq b) \leq c_{12} \alpha \exp(-c_{13} \frac{b}{\alpha})$$

We define a special solution $(\xi, \zeta, \gamma)$ as follows:

$$\xi(x) := I(w_{h^*} \cdot x \geq \frac{r}{c_{11} c_{13}} \ln \frac{c_{12} r}{c_{11} \eta})$$

$$\zeta(x) := I(w_{h^*} \cdot x \leq -\frac{r}{c_{11} c_{13}} \ln \frac{c_{12} r}{c_{11} \eta})$$

$$\gamma(x) := I(|w_{h^*} \cdot x| \leq \frac{r}{c_{11} c_{13}} \ln \frac{c_{12} r}{c_{11} \eta})$$

Then it can be checked that for all $h \in B_D(h^*, r)$,

$$\mathbb{E}[I(h(x) = +1) \zeta(x) + I(h(x) = -1) \xi(x)] = \mathbb{P}_D(\text{sign}(w_{h^*} \cdot x) \neq \text{sign}(w_h \cdot x), |w_{h^*} \cdot x| \geq \frac{r}{c_{11} c_{13}} \ln \frac{c_{12} r}{c_{11} \eta}) \leq \eta$$

And by item (a) of Lemma 2 of [4], we have

$$\mathbb{E} \gamma(x) = \mathbb{P}_D(|w_{h^*} \cdot x| \leq \frac{r}{c_{11} c_{13}} \ln \frac{c_{12} r}{c_{11} \eta}) \leq \frac{r}{c_{11} c_{13}} \ln \frac{c_{12} r}{c_{11} \eta}$$

Hence,

$$\phi(r, \eta) \leq \frac{r}{c_{11} c_{13}} \ln \frac{c_{12} r}{c_{11} \eta}$$

$\square$

*Proof.* (of Corollary 1) This is an immediate consequence of Lemma 14 and Theorems 4 and 5 and algebra. $\square$

# F  A Suboptimal Alternative to Algorithm 2

---
**Algorithm 4** An Nonadaptive Algorithm for Label Query Given Target Excess Error
---
1: **Inputs:** Hypothesis set $V$ of VC dimension $d$, Example distribution $\Delta$, Labeling oracle $\mathcal{O}$, target excess error $\tilde{\epsilon}$, target confidence $\tilde{\delta}$.
2: Draw $n = \frac{12288}{\tilde{\epsilon}^2}(d \ln \frac{12288}{\tilde{\epsilon}^2} + \ln \frac{24}{\tilde{\delta}})$ i.i.d examples from $\Delta$; query their labels from $\mathcal{O}$ to get a labelled dataset $S$.
3: Train an ERM classifier $\hat{h} \in V$ over $S$.
4: Define the set $V$ as follows:

$$V_1 = \left\{ h \in V : \text{err}_S(h) \leq \text{err}_S(\hat{h}) + \frac{3\tilde{\epsilon}}{4} \right\}$$

5: **return** $V_1$.
---

It is immediate that we have the following lemma.

**Lemma 15.** *Suppose we run Algorithm 4 with inputs hypothesis set $V$, example distribution $\Delta$, labelling oracle $\mathcal{O}$, target excess error $\tilde{\epsilon}$ and target confidence $\tilde{\delta}$. Then there exists an event $\tilde{E}$, $\mathbb{P}(\tilde{E}) \geq 1 - \tilde{\delta}$, such that on $\tilde{E}$, the set $V_1$ has the following property. (1) If for $h \in \mathcal{H}$, $\text{err}_{\tilde{\Delta}}(h) - \text{err}_{\tilde{\Delta}}(h^*(\tilde{\Delta})) \leq \tilde{\epsilon}/2$, then $h \in V_1$. (2) On the other hand, if $h \in V_1$, then $\text{err}_{\tilde{\Delta}}(h) - \text{err}_{\tilde{\Delta}}(h^*(\tilde{\Delta})) \leq \tilde{\epsilon}$.*

When $\tilde{E}$ happens, we say that Algorithm 4 succeeds.

*Proof.* By Equation (9) of Lemma 8 and because $n = \frac{12288}{\tilde{\epsilon}^2}(d\ln\frac{12288}{\tilde{\epsilon}^2} + \ln\frac{24}{\tilde{\delta}})$, we have for all $h, h' \in \mathcal{H}$,

$$(\text{err}_{\tilde{\Delta}}(h) - \text{err}_{\tilde{\Delta}}(h')) - (\text{err}_S(h) - \text{err}_S(h')) \leq \frac{\tilde{\epsilon}}{4}$$

For the proof of (1), for any $h \in V$, $\text{err}_{\tilde{\Delta}}(h) - \text{err}_{\tilde{\Delta}}(h^*(\tilde{\Delta})) \leq \tilde{\epsilon}/2$, then

$$\text{err}_{\tilde{\Delta}}(h) - \text{err}_{\tilde{\Delta}}(\hat{h}) \leq \tilde{\epsilon}/2$$

Thus

$$\text{err}_S(h) - \text{err}_S(\hat{h}) \leq \frac{3\tilde{\epsilon}}{4}$$

proving $h \in V_1$.
For the proof of (2), for any $h \in V_1$,

$$\text{err}_S(h) - \text{err}_S(h') \leq \frac{3\tilde{\epsilon}}{4}$$

Thus

$$\text{err}_S(h) - \text{err}_S(h^*(\tilde{\Delta})) \leq \frac{3\tilde{\epsilon}}{4}$$

Combining with the fact that $(\text{err}_{\tilde{\Delta}}(h) - \text{err}_{\tilde{\Delta}}(h^*(\tilde{\Delta}))) - (\text{err}_S(h) - \text{err}_S(h^*(\tilde{\Delta}))) \leq \frac{\tilde{\epsilon}}{4}$ we have

$$\text{err}_{\tilde{\Delta}}(h) - \text{err}_{\tilde{\Delta}}(h^*(\tilde{\Delta})) \leq \tilde{\epsilon}$$

$\square$

**Corollary 2.** *Suppose we replace the calls to Algorithm 2 with Algorithm 4 in Algorithm 1, then run it with inputs example oracle $\mathcal{U}$, labelling oracle $\mathcal{O}$, hypothesis class $V$, confidence-rated predictor $P$ of Algorithm 3, target excess error $\epsilon$ and target confidence $\delta$. Then the modified algorithm has a label complexity of*

$$\tilde{O}\left(\sum_{k=1}^{\lceil\log 1/\epsilon\rceil}(d(\frac{\phi(2\nu^*(D) + \epsilon_k, \epsilon_k/256)}{\epsilon_k})^2)\right)$$

*in the agnostic case and*

$$\tilde{O}\left(\sum_{k=1}^{\lceil\log 1/\epsilon\rceil} d(\frac{\phi(C_0\epsilon_k^{\frac{1}{\kappa}}, \frac{\epsilon_k}{256})}{\epsilon_k^{\frac{1}{\kappa}}})^2\epsilon_k^{\frac{2}{\kappa}-2}\right)$$

*under $(C_0, \kappa)$-Tsybakov Noise Condition.*

Under denseness assumption, by Lemma 17, we have $\phi(r, \eta) \geq r - 2\eta$, the label complexity bounds given by Corollary 2 is always no better than the ones given by Theorem 4 and 5.

*Proof.* (Sketch) Define event

$E_a = \{$For all $k = 1, 2, \ldots, k_0$: Equations (11), (12), (13), (14), (15) hold for $\tilde{U}_k$ with confidence $\delta_k/2$, and Algorithm 4 succeeds with inputs hypothesis set $V = V_k$, example distribution $\Delta = \Gamma_k$, labelling oracle $\mathcal{O}$, target excess error $\tilde{\epsilon} = \frac{\epsilon_k}{8\phi_k}$ and target confidence $\tilde{\delta} = \frac{\delta_k}{2}\}$.

Cleary, $\mathbb{P}(E_a) \geq 1 - \delta$. On the event $E_a$, there exists an absolute constant $c_{13} > 0$, such that the number of examples queried in interation $k$ is

$$m_k \leq c_{13}(d\ln\frac{8\phi_k}{\epsilon_k} + \ln\frac{2}{\delta})(\frac{8\phi_k}{\epsilon_k})^2$$

Combining it with Equation (15) of Lemma 10

$$\phi_k \leq \mathbf{\Phi}_D(V_k, \frac{\epsilon_k}{128}) + \frac{\epsilon_k}{256}$$

we have

$$m_k \leq O\left((d\ln\frac{\mathbf{\Phi}_D(V_k, \frac{\epsilon_k}{128}) + \frac{\epsilon_k}{256}}{\epsilon_k} + \ln\frac{2}{\delta_k})(\frac{\mathbf{\Phi}_D(V_k, \frac{\epsilon_k}{128}) + \frac{\epsilon_k}{256}}{\epsilon_k})^2\right)$$

The rest of the proof follows from Lemma 18 and denseness assumption, along with algebra. $\square$

# G   Proofs of Concentration Lemmas

*Proof.* (of Lemma 9) We begin by observing that:

$$\mathrm{err}_{\tilde{U}_k}(h) = \frac{1}{n_k}\sum_{i=1}^{n_k}[\mathbb{P}_D(Y=+1|X=x_i)I(h(x_i)=-1) + \mathbb{P}_D(Y=-1|X=x_i)I(h(x_i)=+1)]$$

Moreover, $\max(\mathcal{S}(\{I(h(x)=1, h \in \mathcal{H})\}, n), \mathcal{S}(\{I(h(x)=-1, h \in \mathcal{H})\}, n)) \leq (\frac{en}{d})^d$. Combining this fact with Lemma 16, the following equations hold simultaneously with probability $1-\delta_k/6$:

$$\left|\frac{1}{n_k}\sum_{i=1}^{n_k}\mathbb{P}_D(Y=+1|X=x_i)I(h(x_i)=-1) - \mathbb{P}_D(h(x)=-1, y=+1)\right| \leq \sqrt{\frac{16(d\ln\frac{en_k}{d}+\ln\frac{24}{\delta_k})}{n_k}} \leq \frac{\epsilon_k}{128}$$

$$\left|\frac{1}{n_k}\sum_{i=1}^{n_k}\mathbb{P}_D(Y=-1|X=x_i)I(h(x_i)=+1) - \mathbb{P}_D(h(x)=+1, y=-1)\right| \leq \sqrt{\frac{16(d\ln\frac{en_k}{d}+\ln\frac{24}{\delta_k})}{n_k}} \leq \frac{\epsilon_k}{128}$$

Thus Equation (11) holds with probability $1-\delta_k/6$. Moreover, we observe that Equation (11) implies Equation (12). To show Equation (13), we observe that by Lemma 8, with probability $1-\delta_k/12$,

$$|\rho_D(h, h') - \rho_{\tilde{U}_k}(h, h')| = |\rho_D(h, h') - \rho_{S_k}(h, h')| \leq 2\sqrt{\sigma(n_k, \delta_k/12)} \leq \frac{\epsilon_k}{64}$$

Thus, Equation (13) holds with probability $\geq 1-\delta_k/12$. By union bound, with probability $1-\delta_k/4$, Equations (11), (12), and (13) hold simultaneously.  □

*Proof.* (of Lemma 10) (1) Given a confidence-rated predictor with inputs hypothesis set $V_k$, unlabelled data $U_k$, and error bound $\epsilon_k/64$, the outputs $\{(\xi_{k,i}, \zeta_{k,i}, \gamma_{k,i})\}_{i=1}^{n_k}$ must satisfy that for all $h, h' \in V_k$,

$$\frac{1}{n_k}\sum_{i=1}^{n_k}[I(h(x_{k,i})=-1)\xi_{k,i} + I(h(x_{k,i})=+1)\zeta_{k,i}] \leq \frac{\epsilon_k}{64}$$

$$\frac{1}{n_k}\sum_{i=1}^{n_k}[I(h'(x_{k,i})=-1)\xi_{k,i} + I(h'(x_{k,i})=+1)\zeta_{k,i}] \leq \frac{\epsilon_k}{64}$$

Since $I(h(x) \neq h'(x)) \leq \min(I(h(x)=-1)+I(h'(x)=-1), I(h(x)=+1)+I(h'(x)=+1))$, adding up the two inequalities above, we get

$$\frac{1}{n_k}\sum_{i=1}^{n_k}[I(h(x_{k,i}) \neq h'(x_{k,i}))(\xi_{k,i}+\zeta_{k,i})] \leq \frac{\epsilon_k}{32}$$

That is,

$$\frac{1}{n_k}\sum_{i=1}^{n_k}[I(h(x_{k,i}) \neq h'(x_{k,i}))(1-\gamma_{k,i})] \leq \frac{\epsilon_k}{32}$$

(2) By definition of $\Phi_D(V, \eta)$, there exist nonnegative functions $\xi, \zeta, \gamma$ such that $\xi(x) + \zeta(x) + \gamma(x) \equiv 1$, $\mathbb{E}_D[\gamma(x)] = \Phi_D(V_k, \epsilon_k/128)$ and for all $h \in V_k$,

$$\mathbb{E}_D[\xi(x)I(h(x)=-1) + \zeta(x)I(h(x)=+1)] \leq \frac{\epsilon_k}{128}$$

Consider the linear progam in Algorithm 3 with inputs hypothesis set $V_k$, unlabelled data $U_k$, and error bound $\epsilon_k/64$. We consider the following special (but possibly non-optimal) solution for this LP: $\xi_{k,i} = \xi(z_{k,i}), \zeta_{k,i} = \zeta(z_{k,i}), \gamma_{k,i} = \gamma(z_{k,i})$. We will now show that this solution is feasible and has coverage $\Phi_D(V_k, \epsilon_k/128)$ plus $O(\epsilon_k)$ with high probability.
Observe that $\max(\mathcal{S}(\{I(h(x)=1, h \in \mathcal{H})\}, n), \mathcal{S}(\{I(h(x)=-1, h \in \mathcal{H})\}, n)) \leq (\frac{en}{d})^d$. Therefore, from Lemma 16 and the union bound, with probability $1-\delta_k/4$, the following hold simultaneously for all $h \in \mathcal{H}$:

$$\left|\frac{1}{n_k}\sum_{i=1}^{n_k}\gamma(z_{k,i}) - \mathbb{E}_D\gamma(x)\right| \leq \sqrt{\frac{\ln\frac{2}{\delta_k}}{2n_k}} \leq \frac{\epsilon_k}{256} \tag{33}$$

$$\left|\frac{1}{n_k}\sum_{i=1}^{n_k}\xi(z_{k,i})I(h(z_{k,i})=-1)-\mathbb{E}_D[\xi(x)I(h(x)=-1)]\right|\leq\sqrt{\frac{8(d\ln\frac{en_k}{d}+\ln\frac{24}{\delta_k})}{n_k}}\leq\frac{\epsilon_k}{256}\quad(34)$$

$$\left|\frac{1}{n_k}\sum_{i=1}^{n_k}\zeta(z_{k,i})I(h(z_{k,i})=+1)-\mathbb{E}_D[\zeta(x)I(h(x)=+1)]\right|\leq\sqrt{\frac{8(d\ln\frac{en_k}{d}+\ln\frac{24}{\delta_k})}{n_k}}\leq\frac{\epsilon_k}{256}$$
$$(35)$$

Adding up Equations (34) and (35),

$$\left|\frac{1}{n_k}\sum_{i=1}^{n_k}[\zeta(x_i)I(h(x_i)=+1)+\xi(x_i)I(h(x_i)=-1)]-\mathbb{E}_D[\xi(x)I(h(x)=-1)+\zeta(x)I(h(x)=+1))]\right|\leq\frac{\epsilon_k}{128}$$

Thus $\{(\xi(z_{k,i}),\zeta(z_{k,i})\}_{i=1}^{n_k}$ is a feasible solution of the linear program of Algorithm 3. Also, by Equation (33), $\frac{1}{n_k}\sum_{i=1}^{n_k}\gamma(z_{k,i})\leq\Phi_D(V_k,\frac{\epsilon_k}{128})+\frac{\epsilon_k}{64}$. Thus, the outputs $\{(\xi_{k,i},\zeta_{k,i},\gamma_{k,i})\}_{i=1}^{n_k}$ of the linear program in Algorithm 3 satisfy

$$\phi_k=\frac{1}{n_k}\sum_{i=1}^{n_k}\gamma_{k,i}\leq\frac{1}{n_k}\sum_{i=1}^{n_k}\gamma(z_{k,i})\leq\Phi_D(V_k,\frac{\epsilon_k}{128})+\frac{\epsilon_k}{256}$$

due to their optimality. $\qquad\square$

**Lemma 16.** *Pick any $n\geq1$, $\delta\in(0,1)$, a family $\mathcal{F}$ of functions $f:\mathcal{Z}\to\{0,1\}$, a fixed weighting function $w:\mathcal{Z}\to[0,1]$. Let $S_n$ be a set of $n$ iid copies of $Z$. The following holds with probability at least $1-\delta$:*

$$\left|\frac{1}{n}\sum_{i=1}^{n}w(z_i)f(z_i)-\mathbb{E}[w(z)f(z)]\right|\leq\sqrt{\frac{16(\ln\mathcal{S}(\mathcal{F},n)+\ln\frac{2}{\delta})}{n}}$$

*where $\mathcal{S}(\mathcal{F},n)=\max_{z_1,\ldots,z_n\in\mathcal{Z}}|\{(f(z_1),\ldots,f(z_n)):f\in\mathcal{F}\}|$ is the growth function of $\mathcal{F}$.*

*Proof.* The proof is fairly standard, and follows immediately from the proof of additive VC bounds. With probability $1-\delta$,

$$\sup_{f\in\mathcal{F}}\left|\frac{1}{n}\sum_{i=1}^{n}w(z_i)f(z_i)-\mathbb{E}w(z)f(z)\right|$$

$$\leq\quad\mathbb{E}_{S\sim D^n}\sup_{f\in\mathcal{F}}\left|\frac{1}{n}\sum_{i=1}^{n}w(z_i)f(z_i)-\mathbb{E}w(z)f(z)\right|+\sqrt{\frac{2\ln\frac{1}{\delta}}{n}}$$

$$\leq\quad\mathbb{E}_{S\sim D^n,S'\sim D^n}\sup_{f\in\mathcal{F}}\left|\frac{1}{n}\sum_{i=1}^{n}(w(z_i)f(z_i)-w(z_i')f(z_i'))\right|+\sqrt{\frac{2\ln\frac{1}{\delta}}{n}}$$

$$\leq\quad\mathbb{E}_{S\sim D^n,S'\sim D^n,\sigma\sim U(\{-1,+1\}^n)}\sup_{f\in\mathcal{F}}\left|\frac{1}{n}\sum_{i=1}^{n}\sigma_i(w(z_i)f(z_i)-w(z_i')f(z_i'))\right|+\sqrt{\frac{2\ln\frac{1}{\delta}}{n}}$$

$$\leq\quad2\mathbb{E}_{S\sim D^n,\sigma\sim U(\{-1,+1\}^n)}\sup_{f\in\mathcal{F}}\left|\frac{1}{n}\sum_{i=1}^{n}\sigma_iw(z_i)f(z_i)\right|+\sqrt{\frac{2\ln\frac{1}{\delta}}{n}}$$

$$\leq\quad2\sqrt{\frac{2\ln(2\mathcal{S}(\mathcal{F},n))}{n}}+\sqrt{\frac{2\ln\frac{1}{\delta}}{n}}\leq\sqrt{\frac{16(\ln\mathcal{S}(\mathcal{F},n)+\ln\frac{2}{\delta})}{n}}$$

Where the first inequality is by McDiarmid's Lemma; the second inequality follows from Jensen's Inequality; the third inequality follows from symmetry; the fourth inequality follows from $|A+B|\leq|A|+|B|$; the fifth inequality follows from Massart's Finite Lemma. $\qquad\square$

**Lemma 17.** *Let $0<2\eta\leq r\leq1$. Given a hypothesis set $V$ and data distribution $D$ over $\mathcal{X}\times\mathcal{Y}$, if there exist $h_1,h_2\in V$ such that $\rho_D(h_1,h_2)\geq r$, then $\Phi_D(V,\eta)\geq r-2\eta$.*

*Proof.* Let $(\xi, \zeta, \gamma)$ be a triple of functions from $\mathcal{X}$ to $\mathrm{R}^3$ satisfying the following conditions: $\xi, \zeta, \gamma \geq 0$, $\xi + \zeta + \gamma \equiv 1$, and for all $h \in V$,

$$\mathbb{E}_D[\xi(x)I(h(x) = +1) + \zeta(x)I(h(x) = -1)] \leq \eta$$

Then, in particular, we have:

$$\mathbb{E}_D[\xi(x)I(h_1(x) = +1) + \zeta(x)I(h_1(x) = -1)] \leq \eta$$

$$\mathbb{E}_D[\xi(x)I(h_1(x) = +1) + \zeta(x)I(h_2(x) = -1)] \leq \eta$$

Thus, by $I(h_1(x) \neq h_2(x)) \leq \min(I(h_1(x) = -1) + I(h_1(x) = -1), I(h_2(x) = +1) + I(h_2(x) = +1))$, adding the two inequalities up,

$$\mathbb{E}_D[(\xi(x) + \zeta(x))I(h_1(x) \neq h_2(x))] \leq 2\eta$$

Since

$$\rho_D(h_1, h_2) = \mathbb{E}_D I(h_1(x) \neq h_2(x)) \geq r$$

We have

$$\mathbb{E}_D[\gamma(x)I(h_1(x) \neq h_2(x))] = \mathbb{E}_D[(1 - \xi(x) - \zeta(x))I(h_1(x) \neq h_2(x))] \geq r - 2\eta$$

Thus,

$$\mathbb{E}_D[\gamma(x)] \geq \mathbb{E}_D[\gamma(x)I(h_1(x) \neq h_2(x))] \geq r - 2\eta$$

Hence $\mathbf{\Phi}_D(V, \eta) \geq r - 2\eta$. $\qquad\square$

**Lemma 18.** *Given hypothesis set $V$ and data distribution $D$ over $\mathcal{X} \times \mathcal{Y}$, $0 < \lambda < \eta < 1$, if there exist $h_1, h_2 \in V$ such that $\rho_D(h_1, h_2) \geq 2\eta - \lambda$, then $\mathbf{\Phi}_D(V, \eta) + \lambda \leq \mathbf{\Phi}_D(V, \eta - \lambda)$.*

*Proof.* Suppose $(\xi_1, \zeta_1, \gamma_1)$ are nonnegative functions satisfying $\xi_1 + \zeta_1 + \gamma_1 \equiv 1$, and for all $h \in V$, $\mathbb{E}_D[\zeta_1(x)I(h(x) = +1) + \xi_1(x)I(h(x) = -1)] \leq \eta - \lambda$, and $\mathbb{E}_D \gamma_1(x) = \mathbf{\Phi}_D(V, \eta - \lambda)$. Notice by Lemma 17, $\mathbf{\Phi}_D(V, \eta - \lambda) \geq 2\eta - \lambda - 2(\eta - \lambda) = \lambda$.

Then we pick nonnegative functions $(\xi_2, \zeta_2, \gamma_2)$ as follows. Let $\xi_2 = \xi_1$, $\gamma_2 = (1 - \frac{\lambda}{\mathbf{\Phi}_D(V, \eta - \lambda)})\gamma_1$, and $\zeta_2 = 1 - \xi_2 - \gamma_2$. It is immediate that $(\xi_2, \zeta_2, \gamma_2)$ is a valid confidence rated predictor and $\zeta_2 \geq \zeta_1$, $\gamma_2 \leq \gamma_1$, $\mathbb{E}_D \gamma_2(x) = \mathbf{\Phi}_D(V, \eta - \lambda) - \lambda$. It can be readily checked that the confidence rated predictor $(\xi_2, \zeta_2, \gamma_2)$ has error guarantee $\eta$, specifically:

$$
\begin{aligned}
& \mathbb{E}_D[\zeta_2(x)I(h(x) = +1) + \xi_2(x)I(h(x) = -1)] \\
\leq\ & \mathbb{E}_D[(\zeta_2(x) - \zeta_1(x))I(h(x) = +1) + (\xi_2(x) - \xi_1(x))I(h(x) = -1)] + \eta - \lambda \\
\leq\ & \mathbb{E}_D[(\zeta_2(x) - \zeta_1(x)) + (\xi_2(x) - \xi_1(x))] + \eta - \lambda \\
\leq\ & \lambda + \eta - \lambda = \eta
\end{aligned}
$$

Thus, $\mathbf{\Phi}_D(V, \eta)$, which is the minimum abstention probability of a confidence-rated predictor with error guarantee $\eta$ with respect to hypothesis set $V$ and data distribution $D$, is at most $\mathbf{\Phi}_D(V, \eta - \lambda) - \lambda$. $\qquad\square$

## H Detailed Derivation of Label Complexity Bounds

### H.1 Agnostic

**Proposition 1.** *In agnostic case, the label complexity of Algorithm 1 is at most*

$$\tilde{O}\left(\sup_{k \leq \lceil \log(1/\epsilon) \rceil} \frac{\phi(2\nu^*(D) + \epsilon_k, \epsilon_k/256)}{2\nu^*(D) + \epsilon_k}\left(d\frac{\nu^*(D)^2}{\epsilon^2}\ln\frac{1}{\epsilon} + d\ln^2\frac{1}{\epsilon}\right)\right),$$

*where the $\tilde{O}$ notation hides factors logarithmic in $1/\delta$.*

*Proof.* Applying Theorem 5, the total number of labels queried is at most:

$$c_4 \sum_{k=1}^{\lceil \log\frac{1}{\epsilon} \rceil}\left(d\ln\frac{\phi(2\nu^*(D) + \epsilon_k, \epsilon_k/256)}{\epsilon_k} + \ln\left(\frac{\lceil \log(1/\epsilon)\rceil - k + 1}{\delta}\right)\right)\frac{\phi(2\nu^*(D) + \epsilon_k, \epsilon_k/256)}{\epsilon_k}\left(1 + \frac{\nu^*(D)}{\epsilon_k}\right)$$

Using the fact that $\phi(2\nu^*(D) + \epsilon_k, \epsilon_k/256) \leq 1$, this is

$$c_4 \sum_{k=1}^{\lceil \log \frac{1}{\epsilon} \rceil} (d \ln \frac{\phi(2\nu^*(D) + \epsilon_k, \epsilon_k/256)}{\epsilon_k} + \ln(\frac{\lceil \log(1/\epsilon) \rceil - k + 1}{\delta})) \frac{\phi(2\nu^*(D) + \epsilon_k, \epsilon_k/256)}{\epsilon_k} (1 + \frac{\nu^*(D)}{\epsilon_k})$$

$$= \tilde{O} \left( \sum_{k=1}^{\lceil \log \frac{1}{\epsilon} \rceil} (d \ln \frac{\phi(2\nu^*(D) + \epsilon_k, \epsilon_k/256)}{\epsilon_k} + \ln \log(1/\epsilon)) \frac{\phi(2\nu^*(D) + \epsilon_k, \epsilon_k/256)}{2\nu + \epsilon_k} (1 + \frac{\nu^*(D)^2}{\epsilon_k^2}) \right)$$

$$\leq \tilde{O} \left( \sup_{k \leq \lceil \log(1/\epsilon) \rceil} \frac{\phi(2\nu^*(D) + \epsilon_k, \epsilon_k/256)}{2\nu^*(D) + \epsilon_k} \sum_{k=1}^{\lceil \log \frac{1}{\epsilon} \rceil} (1 + \frac{\nu^*(D)^2}{\epsilon_k^2})(d \ln \frac{1}{\epsilon} + \ln \ln \frac{1}{\epsilon}) \right)$$

$$\leq \tilde{O} \left( \sup_{k \leq \lceil \log(1/\epsilon) \rceil} \frac{\phi(2\nu^*(D) + \epsilon_k, \epsilon_k/256)}{2\nu^*(D) + \epsilon_k} (d \frac{\nu^*(D)^2}{\epsilon^2} \ln \frac{1}{\epsilon} + d \ln^2 \frac{1}{\epsilon}) \right),$$

where the last line follows as $\epsilon_k$ is geometrically decreasing. $\qquad\square$

## H.2 Tsybakov Noise Condition with $\kappa > 1$

**Proposition 2.** *Suppose the hypothesis class $\mathcal{H}$ and the data distribution $D$ satisfies $(C_0, \kappa)$-Tsybakov Noise Condition with $\kappa > 1$. Then the label complexity of Algorithm 1 is at most*

$$\tilde{O}(\sup_{k \leq \lceil \log(1/\epsilon) \rceil} \frac{\phi(C_0 \epsilon_k^{\frac{1}{\kappa}}, \frac{\epsilon_k}{256})}{\epsilon_k^{\frac{1}{\kappa}}} \epsilon^{\frac{2}{\kappa} - 2} d \ln \frac{1}{\epsilon}),$$

*where the $\tilde{O}$ notation hides factors logarithmic in $1/\delta$.*

*Proof.* Applying Theorem 5, the total number of labels queried is at most:

$$c_5 \sum_{k=1}^{\lceil \log \frac{1}{\epsilon} \rceil} (d \ln(\phi(C_0 \epsilon_k^{\frac{1}{\kappa}}, \frac{\epsilon_k}{256}) \epsilon_k^{\frac{1}{\kappa} - 2}) + \ln(\frac{k_0 - k + 1}{\delta})) \phi(C_0 \epsilon_k^{\frac{1}{\kappa}}, \frac{\epsilon_k}{256}) \epsilon_k^{\frac{1}{\kappa} - 2}$$

Using the fact that $\phi(C_0 \epsilon_k^{\frac{1}{\kappa}}, \frac{\epsilon_k}{256}) \leq 1$, we get

$$c_5 \sum_{k=1}^{\lceil \log \frac{1}{\epsilon} \rceil} (d \ln(\phi(C_0 \epsilon_k^{\frac{1}{\kappa}}, \frac{\epsilon_k}{256}) \epsilon_k^{\frac{1}{\kappa} - 2}) + \ln(\frac{k_0 - k + 1}{\delta})) \phi(C_0 \epsilon_k^{\frac{1}{\kappa}}, \frac{\epsilon_k}{256}) \epsilon_k^{\frac{1}{\kappa} - 2}$$

$$\leq \tilde{O} \left( \sup_{k \leq \lceil \log(1/\epsilon) \rceil} \frac{\phi(C_0 \epsilon_k^{\frac{1}{\kappa}}, \frac{\epsilon_k}{256})}{\epsilon_k^{\frac{1}{\kappa}}} \sum_{k=1}^{\lceil \log \frac{1}{\epsilon} \rceil} \epsilon_k^{\frac{2}{\kappa} - 2} d \ln \frac{1}{\epsilon} \right)$$

$$\leq \tilde{O} \left( \sup_{k \leq \lceil \log(1/\epsilon) \rceil} \frac{\phi(C_0 \epsilon_k^{\frac{1}{\kappa}}, \frac{\epsilon_k}{256})}{\epsilon_k^{\frac{1}{\kappa}}} \epsilon^{\frac{2}{\kappa} - 2} d \ln \frac{1}{\epsilon} \right)$$

$\qquad\square$

## H.3 Fully Agnostic, Linear Classification of Log-Concave Distribution

We show in this subsection that in agnostic case, if $\mathcal{H}$ is the class of homogeneous linear classifiers in $\mathbb{R}^d$, $D_{\mathcal{X}}$ is isotropic log-concave in $\mathbb{R}^d$, then, our label complexity bound is at most

$$O(\ln \frac{\epsilon + \nu^*(D)}{\epsilon} (\ln \frac{1}{\epsilon} + \frac{\nu^*(D)^2}{\epsilon^2})(d \ln \frac{\epsilon + \nu^*(D)}{\epsilon} + \ln \frac{1}{\delta}) + \ln \frac{1}{\epsilon} \ln \frac{\epsilon + \nu^*(D)}{\epsilon} \ln \ln \frac{1}{\epsilon})$$

Recall by Lemma 14, we have $\phi(2\nu^*(D) + \epsilon_k, \epsilon_k/256) \leq C(\nu^*(D) + \epsilon_k) \ln \frac{\nu^*(D) + \epsilon_k}{\epsilon_k}$ for some constant $C > 0$. Applying Theorem 4, the label complexity is

$$O(\sum_{k=1}^{\lceil \log \frac{1}{\epsilon} \rceil} (d \ln(\frac{2\nu^*(D) + \epsilon_k}{\epsilon_k} \ln \frac{2\nu^*(D) + \epsilon_k}{\epsilon_k}) + \ln(\frac{\log(1/\epsilon) - k + 1}{\delta})) \ln \frac{2\nu^*(D) + \epsilon_k}{\epsilon_k} (1 + \frac{\nu^*(D)^2}{\epsilon_k^2}))$$

This can be simplified to (treating $1$ and $\frac{\nu^*(D)^2}{\epsilon_k^2}$ separately)

$$O(\sum_{k=1}^{\lceil \log \frac{1}{\epsilon} \rceil} \ln \frac{\nu^*(D) + \epsilon_k}{\epsilon_k} (d \ln \frac{\nu^*(D) + \epsilon_k}{\epsilon_k} + \ln \frac{k_0 - k + 1}{\delta})$$

$$+ \sum_{k=1}^{\lceil \log \frac{1}{\epsilon} \rceil} \frac{\nu^*(D)^2}{\epsilon_k^2} \ln \frac{\nu^*(D) + \epsilon_k}{\epsilon_k} (d \ln \frac{\nu^*(D) + \epsilon_k}{\epsilon_k} + \ln \frac{k_0 - k + 1}{\delta}))$$

$$\leq O(\ln \frac{1}{\epsilon} \ln \frac{\epsilon + \nu^*(D)}{\epsilon} (d \ln \frac{\epsilon + \nu^*(D)}{\epsilon} + \ln \ln \frac{1}{\epsilon} + \ln \frac{1}{\delta}) + \frac{\nu^*(D)^2}{\epsilon^2} \ln \frac{\epsilon + \nu^*(D)}{\epsilon} (d \ln \frac{\epsilon + \nu^*(D)}{\epsilon} + \ln \frac{1}{\delta}))$$

$$\leq O(\ln \frac{\epsilon + \nu^*(D)}{\epsilon} (\ln \frac{1}{\epsilon} + \frac{\nu^*(D)^2}{\epsilon^2})(d \ln \frac{\epsilon + \nu^*(D)}{\epsilon} + \ln \frac{1}{\delta}) + \ln \frac{1}{\epsilon} \ln \frac{\epsilon + \nu^*(D)}{\epsilon} \ln \ln \frac{1}{\epsilon})$$

### H.4 Tsybakov Noise Conditon with $\kappa > 1$, Log-Concave Distribution

We show in this subsection that under $(C_0, \kappa)$-Tsybakov Noise Condition, if $\mathcal{H}$ is the class of homogeneous linear classifiers in $\mathrm{R}^d$, and $D_{\mathcal{X}}$ is isotropic log-concave in $\mathrm{R}^d$, our label complexity bound is at most

$$O(\epsilon^{\frac{2}{\kappa} - 2} \ln \frac{1}{\epsilon} (d \ln \frac{1}{\epsilon} + \ln \frac{1}{\delta}))$$

Recall by Lemma 14, we have $\phi(C_0 \epsilon_k^{\frac{1}{\kappa}}, \frac{\epsilon_k}{256}) \leq C \epsilon_k^{\frac{1}{\kappa}} \ln \frac{1}{\epsilon_k}$ for some constant $C > 0$. Applying Theorem 5, the label complexity is:

$$O(\sum_{k=1}^{\lceil \log \frac{1}{\epsilon} \rceil} (d \ln(\phi(C_0 \epsilon_k^{\frac{1}{\kappa}}, \frac{\epsilon_k}{256}) \epsilon_k^{\frac{1}{\kappa} - 2}) + \ln(\frac{k_0 - k + 1}{\delta})) \phi(C_0 \epsilon_k^{\frac{1}{\kappa}}, \frac{\epsilon_k}{256}) \epsilon_k^{\frac{1}{\kappa} - 2})$$

This can be simplified to :

$$O(\sum_{k=1}^{\lceil \log \frac{1}{\epsilon} \rceil} (d \ln(\epsilon_k^{\frac{2}{\kappa} - 2} \ln \frac{1}{\epsilon_k}) + \ln(\frac{k_0 - k + 1}{\delta})) \epsilon_k^{\frac{2}{\kappa} - 2} \ln \frac{1}{\epsilon_k})$$

$$\leq O((\sum_{k=1}^{\lceil \log \frac{1}{\epsilon} \rceil} \epsilon_k^{\frac{2}{\kappa} - 2}) \ln \frac{1}{\epsilon} (d \ln \frac{1}{\epsilon} + \ln \frac{1}{\delta}))$$

$$\leq O(\epsilon^{\frac{2}{\kappa} - 2} \ln \frac{1}{\epsilon} (d \ln \frac{1}{\epsilon} + \ln \frac{1}{\delta}))$$