[Reviews · NeurIPS 2014]

Submitted by Assigned_Reviewer_31

In this paper, the authors proposed a new active learning algorithm, which avoids the disagreement coefficient in the argument and label complexity. The proposed label complexity is also slight tighter than existing algorithm. In details, the authors first show a general connection between confidence-rated predictors and active learning. Given a confidence-rated predictor with guaranteed error, the authors show how to use it to construct an active label query algorithm consistent in the agnostic setting. A novel confidence-rated predictor with guaranteed error that applies to any general classification problem is also proposed. They show that this predictor is optimal in the realizable case, in the sense that it has the lowest abstention rate out of all predictors that guarantee a certain error. Moreover, they show how to extend our predictor to the agnostic setting.

Detailed comments:
1. The last step in the proof of Lemma 14 is wrong. However, a somewhat similar bound as that in Lemma 14 can be proved. Please correct it.
2. The setting of sigma(n,\delta) in line 233 seems inconsistent with the proof in the supplementary material. This again comes from the incorrect bound derived in Lemma 14. Please double check it.
3. The notations of \tilde{U}_k and S_k are a bit confusing, and seem to be used interchangeably in the supplementary.

Overall, this is a strong paper, clearly above the NIPS acceptance threshold.
Summary: This is a strong paper, although there are minor mistakes in its proof.

Submitted by Assigned_Reviewer_34

This paper proposes a new algorithm and analysis for agnostic active learning. Active learning is a very lively area of research and, in recent years, various directions for provably successful agnostic active learners have been developed (as outlined nicely in this contributions). In particular, this paper provides a consistent (for any hypothesis class) active learner that has better label complexity than disagreement-based active learners (which has been an open challenge for some years now).

The proposed method is reminiscent of the margin-based active learning algorithms. However, abstracting from that method, the authors bring in a new tool to the development of active learners, namely confidence-rated predictors (roughly, learners that are allowed to abstain from prediction on a portion of a prespecified set of unlabeled data). This new connection may be of independent interest. Using confidence-rated predictors, this work designs an active learner and analyses its sample complexity in terms of the behavior of the confidence-rated predictors and the error-parameter epsilon. The bounds are instantiated for two cases, that have been shown to allow label improvements through active learning before: learning under the Tsybakov noise condition and learning linear classifiers under log-concave distributions.

The main drawback of this work is that it still remains somewhat unclear, under which scenarios the proposed algorithm has better label complexity than passive learning (which should be the primary objective; rather than improving over the label complexity of disagreement-based active learners that suffer from the same drawback..). The two case studies of instantiating the bounds mainly recover previously obtained label complexity bounds. The authors should make this (discussion of improving over passive learning) more transparent.

Nevertheless, this is an interesting (and well written) study, that contains sufficient novel contributions to be of value to the ML research community and should be published.

Minor comments:

Why switch between \Pi and D for the data distribution in Section 2.1?

I found the use of symbol 0 for abstention confusing at first read, since 0 could also be binary label...
Summary: This paper proposes a new algorithm and analysis for agnostic active learning.
It brings new tools into the analysis of active learning algorithms and deserves publication.

Submitted by Assigned_Reviewer_36

This work is on general sample complexity analysis of active learning.
The paper proposes a strategy that requests labels in a subset of
the region of disagreement, and thereby obtains sample complexity
guarantees that are sometimes better than those obtained via the
disagreement coefficient analysis of disagreement-based active
learning methods. This essentially represents a generalization of
the margin-based active learning technique of BBZ07. The sample
complexity bounds are general (applicable to any VC class), and
are instantiated for linear separators under distributional
assumptions as an example. Bounds are derived for several noise
models, including the realizable case, agnostic case with a given
noise rate, and Tsybakov noise.

More abstractly, the paper establishes an elegant connection
between the work on margin-based active learning techniques
as refinements of disagreement-based methods, and the
more-recent work of El-Yaniv & Wiener discussing the use of
selective classification methods for selective sampling.
Specifically, El-Yaniv & Wiener have studied the use of perfect
selective classification methods to do active learning in the
realizable case, and argued that the use of an optimal such
method results in the classic CAL active learning algorithm,
which is disagreement-based. From this perspective, the present
work is instead investigating the use of *imperfect* selective
classification methods (which have a nonzero, but controlled,
probability of error on their predictions), and argues that this
approach can recover essentially similar behavior to the existing
margin-based methods for linear separators (as a special case of
this general approach). This appears to be a nice contribution,
and the idea may potentially generate future work.

The two comments I might make are:
1. It would be nice to have an example other than linear separators,
to demonstrate that this general technique has interesting
implications beyond what is already known.
2. I suggest being a little careful about the claims in the paper.
What is shown is that this upper bound on the sample complexity
of this method is sometimes superior to the upper bound known
for disagreement-based methods in terms of the disagreement
coefficient. This is different from showing that the method
sometimes achieves a sample complexity superior to that achieved
by disagreement-based methods.
Summary: This work develops a general method and analysis, which is sometimes better than the
disagreement coefficient analysis, and generalizes the margin-based active learning technique
for linear separators. It makes a substantial contribution to the theoretical active learning
literature, and is likely to generate future work based on these ideas.
Author Feedback
Author rebuttal: We thank the reviewers for their comments and feedback.

Reviewer_36 has pointed out some claims that need to be clarified; we agree and will make our statements more precise in the final version.

Reviewer_31:

Thank you for your detailed feedback, and for pointing out the minor mistake in the last step of the proof of Lemma 14! As a consequence, the definition of \sigma(n,\delta) should change by a factor of 2 to (16/n) * (2d\ln (2en/d) + \ln (24/\delta)), and the choice of n_k and m_k will change by a constant factor to 192(512/\epsilon_k)^2(d\ln 512/epsilon_k + \ln 288/\delta_k) and 1536\phi_k / \epsilon_k (d ln 1536\phi_k / \epsilon_k + \ln 48 / \delta_k). Other than this constant factor change, the main result and conclusions will remain unchanged. This will be fixed in the final version of the paper.